# Flavonoids and Related Members of the Aromatic Polyketide Group in Human Health and Disease: Do They Really Work?

**DOI:** 10.3390/molecules25173846

**Published:** 2020-08-24

**Authors:** Jan Tauchen, Lukáš Huml, Silvie Rimpelova, Michal Jurášek

**Affiliations:** 1Department of Food Science, Faculty of Agrobiology, Food and Natural Resources, Czech University of Life Sciences Prague, Kamýcká 129, Praha 6, 165 00 Praha, Czech Republic; 2Department of Chemistry of Natural Compounds, University of Chemistry and Technology Prague, Technická 5, Prague 6, 166 28 Prague, Czech Republic; lukas.huml@vscht.cz (L.H.); michal.jurasek@gmail.com (M.J.); 3Department of Biochemistry and Microbiology, Faculty of Food and Biochemical Technology, University of Chemistry and Technology Prague, Technická 3, Prague 6, 166 28 Prague, Czech Republic; silvie.rimpelova@vscht.cz

**Keywords:** flavonoids, polyketides, antioxidants, anticancer, clinical significance, dietary supplements, nutrition

## Abstract

Some aromatic polyketides such as dietary flavonoids have gained reputation as miraculous molecules with preeminent beneficial effects on human health, for example, as antioxidants. However, there is little conclusive evidence that dietary flavonoids provide significant leads for developing more effective drugs, as the majority appears to be of negligible medicinal importance. Some aromatic polyketides of limited distribution have shown more interesting medicinal properties and additional research should be focused on them. Combretastatins, analogues of phenoxodiol, hepatoactive kavalactones, and silymarin are showing a considerable promise in the advanced phases of clinical trials for the treatment of various pathologies. If their limitations such as adverse side effects, poor water solubility, and oral inactivity are successfully eliminated, they might be prime candidates for the development of more effective and in some case safer drugs. This review highlights some of the newer compounds, where they are in the new drug pipeline and how researchers are searching for additional likely candidates.

## 1. Introduction

Plants have always played a significant role in human life. They have been used in nearly every aspect of human needs, as food, medicine, spices, beverages, sugar, oil, dyes, clothing, building materials, pesticides and hunting poisons. A plethora of different plants has been used in ritual ceremonies and traditional medicine. The knowledge of the use of medicinal plants provided an important clue that particular plants might contain chemicals with significant health-affecting activities and eventually became the basis of a plant-derived pharmacopeia. The various extracts, potions and formulations in a combination with advances in medicinal chemistry led researchers to the discovery of a large number of drugs, many of which are still in use today. It is estimated that our planet is home to 250,000–300,000 species of higher plants, of which only a relatively small number have been characterized from a phytochemical standpoint with a view towards using them pharmaceutically [1]. Thus, there is still a significant chance of discovering new plant-derived compounds with useful pharmaceutical properties that could eventually lead to clinically effective new drugs. And now, with the development of molecular medicine and the capability to study signalling pathways and drug targets in detail, naturally-derived agents that have previously failed in the clinic might become the focus of renewed interest. The position of natural products as sources of new pharmaceutical agents, especially anticancer drugs, is still very important, as recently reviewed by Newman and Cragg (2016) [2].

It has been estimated that 500,000–600,000 plant-derived compounds have been described, of which about 100,000 have demonstrated promising biological activity that could be of value in the treatment of human diseases [3]. Many chemical groups such as polyketides, cannabinoids and many others derived from plants have been the starting points of successful therapeutics, such as shikimate-derived compounds (Tamiflu^®^, etoposide, teniposide), terpenoids (artemisinin and Taxol^®^), and high molecular weight compounds (ricin and polysaccharides). Certain groups, however, have demonstrated greater medicinal utility than the others. Plant-derived alkaloids have supplied a large share of clinically useful drugs including cocaine, tubocurarine, morphine, quinine, vincristine, vinblastine, ellipticine, camptothecin, and physostigmine [4]. Pharmaceutical interest in alkaloids as important drug compounds remains at a steady rate, while the interest in flavonoids, perhaps the most common example of aromatic polyketides, has dramatically increased over the last decade (Figure 1). In this review, we focus on the reasons behind the enormous interest in flavonoids, despite the fact that the therapeutic efficacy of many of them is highly questionable.

Plant-derived aromatic polyketides are collectively referred to as compounds that combine biosynthetic features from both shikimate and acetate pathways, usually incorporating aromatic rings and polyketide-derived groups on their structures, hence the name. There are other plant-derived polyketides with an aromatic functional group, such as emodin, sennosides, and hypericin. However, in the course of biosynthesis, these structures utilize acetate starter units (e.g., acetyl-CoA, propionyl-CoA) instead of that derived from shikimate metabolism (e.g., *p*-coumaroyl-CoA) and are usually products of type II polyketide synthase (PKS) enzymes (not type III PKS; see further sections for more details). Aromatic polyketides that are discussed in this review are regarded as a subgroup which, from the nomenclature point of view, fall under the larger group of shikimate-derived compounds, and are further divided into styrylpyrones, diarylheptanoids, flavonoids and stilbenes, flavonolignans, and isoflavonoids [4]. Some of these compounds, particularly flavonoids, are present in large amounts in most plant tissues, whereas others have a rather restricted distribution and abundancy. Flavonoids were demonstrated to have very important roles in plant reproduction, defence, and hormonal regulation, but the degree to which they are functionally active in humans remains uncertain.

Since flavonoids are present in fruits, vegetables, spices, beverages, and other common foodstuffs, humans may consume appreciable quantities (>500 mg) of these compounds every day [5]. Flavonoids have been presented by the media and in advertisement and even in some scientific articles as molecules with powerful health-promoting and anti-aging properties. They are viewed as strong antioxidants and free radical scavengers that can prevent the oxidation of lipids, proteins and nucleic acids, which are essential for the maintenance of the basic physiological functions of an organism. Flavonoids have been present in the human diet for centuries and adverse biological effects have generally not been reported. This might raise the question of whether these compounds have any therapeutic benefit because most effective medicines do have side effects. Recent studies seem to be predominantly focused on the widely distributed common members of the aromatic polyketide family like dietary flavonoids (some examples are given in Table 1), which are somewhat non-critically perceived as miracle molecules with broad health-promoting effects. However, those with a much more restricted distribution, such as specific stilbenes, styrylpyrones, flavonolignans, and various isoflavonoids might be more promising as clinically useful therapeutics (Table 2).

This review focused on the clinical data concerning flavonoids and the related aromatic polyketides obtained through an extensive literature review and a search of the relevant books and research articles with the use of Web of Knowledge, SciVerse, Scopus and PubMed databases. In order to highlight particular agents as clinically relevant, special emphasis was placed on their selection by applying the following criteria: what phases of clinical trials have been passed or, for those compounds introduced into medicinal practice, at what stage of clinical development are they. If clinical data were unavailable at the time, studies describing the results from animal or cell culture were taken into account. Besides the clinical efficacy, this review also describes the botanical source of each agent, for which disease it might be of value, its biological activity, proposed mechanism of action, and side-effects.

## 2. Flavonoids and Stilbenes

Flavonoids and stilbenes are derived from a *p*-coumaroyl-CoA starter unit that is extended by the addition of three units of malonyl-CoA. A group of plant enzymes that belongs to the family of type III polyketide synthases (PKS) controls the extension reaction, as well as further modification steps such as cyclisation, enolization, and decarboxylation. The initially formed polyketide may be subsequently folded in two different ways, giving rise to either stilbene (through aldol and ester hydrolysis by stilbene synthase) or naringenin chalcone (via a Claisen-like reaction catalysed by chalcone synthase in the formation of *p*-coumaroylcyclohexantrione). Naringenin-chalcone can then undergo enolization and a Michael-type nucleophilic attack of OH-group onto the α,β-unsaturated ketone (through chalcone isomerase), forming a general 15-carbon flavonoid structure consisting of two phenyl rings (A and B) and a six-membered heterocyclic ring (C) [6,7,8]. Alternatively, the action of chalcone reductase in the early steps of the flavonoid biosynthetic pathway may generate isoliquiritigenin (chalcone), subsequently leading to liquiritigenin (**1**), Figure 2, precursor of many isoflavonoids. It is worth noting, though, that the majority of flavonoid structures are derived from naringenin chalcone. The primary forms of flavanone structures (e.g., naringenin) may give rise to a wide range of variants on this basic skeleton, including flavons, flavonols, dihydroflavonols, catechines, and anthocyanidins [9,10,11]. Each of the named subgroups can undergo a different degree of hydroxylation of the aromatic rings and these groups can be further modified by methylation, prenylation, glycosylation, or dimethylallylation [12]. Some flavonoid structures (such as liquiritigenin) may lose one of the hydroxyl groups. A large range of flavonoid-derived compounds exists with more than 8,000 stillbenoid and flavonoid structures described thus far [13]. Stilbenes and flavonoids are found mainly in vascular plants, primarily in angiosperms. However, some flavonoid structures have been isolated from algae, molluscs, fungi, corals, and bacteria [14]. Animals lack the enzymatic apparatus required for their biosynthesis and obtain these compounds through diet. Stilbene synthase has a much more restricted distribution in the plant kingdom compared to chalcone synthase. This is supported by the fact that nearly every higher plant contains flavonoids, but stilbenes are found in only a few genera. Some flavonoids, however, may undergo specialized conversions and these might also have a rather unique distribution in higher plants (and other organisms) [4].

### 2.1. Role in Plants

Plants produce flavonoids and stilbenes for various purposes. The name flavonoid is derived from the Latin flavus meaning yellow. Flavonoids possess a strong chromophore, producing various colours from white as in luteolin, to the yellow of quercetin, pink of peonidin, or blue/black of cyanidin and delphinidin. Flavonoids were first discovered in flowers and fruits, and solely or together with other compounds like carotenoids are responsible for their colour, which helps in attracting pollinators and in seed dispersal [15]. Flowers and fruits are not the only plant parts that contain flavonoids, however. They have also been isolated from leaves, bark, stems, and roots, in which they chiefly accumulate in vacuoles and cytoplasm, suggesting that they play other roles than just assisting in plant reproduction and gene pool transfer [16]. Flavonoids were shown to protect the plants against excessive UV light, as a defence against bacterial and fungal pathogens, to deter herbivory, and to prevent damage from frost, drought, heat, and heavy metals. These compounds were also shown to mediate both intra-organismal and extra-organismal interactions, including the regulation of hormonal activity such as auxin [17], communication between pollen and stigma, and allelopathic interactions with other plants [18,19,20,21].

### 2.2. Roles in Humans

#### 2.2.1. Antioxidant Activity of Flavonoids and Stilbenes

Flavonoids, especially those consumed in the diet, have become popular as molecules with strong antioxidant potential and beneficial actions in diabetes, cardiovascular disorders, various types of cancer and inflammatory conditions. Although many studies claimed that oxidative stress had a substantial impact on the onset and development of secondary pathology in more than 150 human diseases, not every disease can be linked to oxidative stress. Nevertheless, it has been established that oxidative stress does play a negative role in many human diseases such as the cancer of the stomach, liver, prostate, ovaries and breasts, as well as neurodegenerative diseases, Alzheimer’s and Parkinson’s, inflammatory bowel disease, Crohn’s disease, ulcerative colitis and arthrosclerosis. As thoroughly reviewed by Halliwell (2012) [22], flavonoids as antioxidants have failed in the majority of intervention studies to provide any therapeutic benefit and thus have not passed to further phases of clinical trials, let alone used as antioxidants in medical practice. Interestingly, various studies point to a fact that flavonoids might actually act in a pro-oxidant manner in vivo, rather than as antioxidants (such might be the case of resveratrol [23]). Since the human diet may have been relatively rich in flavonoids, humans must have adapted to this evolutionary pressure. Indeed, any specific cellular transporter that would be responsible for the accumulation of flavonoids in particular tissues have not been found yet [24]. Therefore, it seems that these compounds can be readily excreted from biological matrices such as urine or faeces. Some flavonoids might provide a therapeutic benefit in human diseases through a different biological mechanism than that of antioxidant activity. Flavonoids were found to inhibit various enzymes such as cyclin-dependent kinases. Therefore, the question if flavonoids are effective in the treatment of oxidative stress-related diseases, and whether their effect is provided by an antioxidant mechanism is still being debated. The answer to the first part of the question appears to be “maybe”, but to the second “probably not”. Further enlightenment on this problem requires more future research.

#### 2.2.2. Health Benefits of Common Dietary Flavonoids and Stilbenes via Non-Antioxidant Mechanisms

The evidence to date seems to indicate that flavonoids and stilbenes are not effective as antioxidants in vivo, but that they might provide therapeutic effects via some other mechanism. On the following pages, the most frequently researched compounds are discussed individually (for structures see Figure 2).

Quercetin (**7**), Figure 2, is perhaps the best known of the basic flavonoids. It was originally isolated in 1895 from the bark of the oak, *Quercus tinctoria*; Fagaceae, hence the name. It demonstrated modest anticancer activity by interfering with protein tyrosine kinases [25]. This discovery led to the development of agents with improved anticancer activity such as the Cdk-inhibiting rohitukine-derived drugs, discussed later. The anticancer effect of quercetin was tested in a phase I clinical trial, but it was not advanced to later testing stages, presumably because of the discovery of more effective agents. Quercetin was also found to be an oestrogen receptor agonist similar to isoflavonoids (discussed below in Section 3.4) [26]. The clinical efficacy of quercetin is far from conclusive and needs more testing. Despite these facts, quercetin is marketed as a dietary supplement and some manufacturers claim that it is of benefit in various human diseases, including some cancers such as that of the prostate [27]. As a result, the Food and Drug Administration (FDA) issued warning letters to some manufacturers to cease making unauthorized health claims [28].

Kaempferol (**6**), Figure 2, seems to have properties similar to quercetin. It showed a relatively broad spectrum of anticancer activity in vitro and in animal models of breast, ovarian, prostate, gastric, pancreatic, and lung cancer. In some types of cancer, the anticancer effects of kaempferol were believed to be associated with its ability to interact with oestrogenic receptors (compare with isoflavonoids discussed later in this text). It was also suggested to be involved in the regulation of the cell cycle, suppression of metastasis, inhibition of angiogenesis, and the induction of apoptosis [29]. Kaempferol also inhibits fatty acid amide hydrolase (FAAH), an enzyme responsible for the breakdown of various endocannabinoids (such as anandamide), sleep-inducing lipids (oleamides), and various members of the transient receptor potential of calcium channels (e.g., *N*-acyltaurines) [30]. This activity might be useful in combating anxiety, pain, and sleep disorders. Various studies also claimed that there was a link between the consumption of kaempferol-rich foods such as certain fruits, tea, and some vegetables from the Brassicaceae family, and a reduced risk of type 2 diabetes. A kaempferol derivative, 6-methoxykaempferol-3-*O*-β-D-robinobioside, inhibited aldose reductase, which is assumed to be associated with diabetes complications [31]. Various other biological functions were reported for kaempferol, including anti-inflammatory, antibacterial, neuroprotective, anti-osteoporotic, anxiolytic, analgesic and anti-allergic activities [32]. None of these claims were supported by clinical trials, however.

Taxifolin (**9**), Figure 2, is a dihydro variant of quercetin that possesses some of the properties of the structurally related flavonoids, quercetin and kaempferol. It shows relatively strong anticancer activity in ovarian cancer through the inhibition of the cell cycle, angiogenesis, and VEGF (vascular endothelial growth factor) expression [33]. Other mechanisms of the anticancer action of taxifolin were also proposed, such as the ability to inhibit fatty acid synthase [34]. Taxifolin has also been shown to stimulate fibril formation and promote the stabilization of fibrillar forms of collagen. This property was also observed for other flavonoids such as kaempferol [35]. Taxifolin and luteolin also inhibited melanogenesis in murine melanoma cells [36], which suggests that these compounds might have skin-bleaching activity in vivo. Similar to other flavonoids like catechin and apigenin, taxifolin appears to have some mild antagonistic activity towards opioid receptors [37]. No clinical data are available to support the use of taxifolin in the treatment of any particular disease.

Naringenin (**2**), Figure 2, has been quite extensively studied as a possible treatment for neurodegenerative diseases. It enhances memory and ameliorates the characteristic pathologies in an animal model of Alzheimer’s disease by binding to CRMP2, collapsin response mediator protein type 2 [38]. Evidence was also reported that naringenin might be of benefit in Parkinson’s disease by reducing α-synuclein pathology and neuroinflammation both in vitro and in animal models [39,40]. Naringenin is also currently under investigation as an antiviral agent against hepatitis C virus. Additionally, there is some evidence that naringenin can inhibit some human cytochrome P-450 enzymes, which may interfere with the metabolism of certain drugs. The consumption of citrus fruits and tomatoes that are rich in naringenin may produce adverse effects and possibly affect some medications.

Apigenin (**4**), Figure 2, was shown to interact with numerous receptors, including benzodiazepine, γ-aminobutyric acid (GABA), adenosine, opioid, PPAR-γ (peroxisome proliferator-activated receptors), and *N*-methyl-D-aspartate (NMDA). It is an activator of the monoamine transporter system and was also found to have selective anticancer activity through interference with the cell cycle and the regulation of protease production [41]. Similar to kaempferol, apigenin also inhibits FAAH [30], a feature associated with anxiolytic properties. Recently, it was also claimed that apigenin was able to cross the blood–brain barrier [42]; which would put it into the category of a potential therapeutic treatment for Alzheimer’s disease. The evidence for apigenin’s pharmacological activity, however, has so far only come from in vitro experiments and animal models [43]. Like naringenin, apigenin appears to be a potential inhibitor of cytochrome P-450 enzymes such as CYP2C9 and thus may potentiate the action of some drugs [44].

Myricetin (**18**), Figure 3, is commonly found in foods such as fruits, berries, nuts, vegetables, and tea, but has also been isolated from medicinal plants in the Myricaceae family like *Myrica rubra*. In contrast to its known antioxidant properties, myricetin also appeared to be a possible pro-oxidant, especially in the presence of specific ions such as Fe^2+^ (compare to resveratrol below). Thus, there have been concerns that myricetin may cause DNA damage. In contrast, some studies showed that myricetin could protect cells from mutagenesis and tumorigenesis by environmental pollutants like polycyclic aromatic hydrocarbons, but the mechanism of its protective action was not established. Myricetin is also considered to possess powerful anti-inflammatory activity, although the evidence is mostly from animal models and in vitro studies. Various mechanisms for its action have been proposed including the inhibition of cytokines, IL-12, IL-1β and TNFα or the inhibition of cyclooxygenase and lipoxygenase enzymes, COX-1, COX -2, and 5-LOX [45]. Myricetin was also tested in clinical trials as an antidiabetic agent, although with equivocal results [46]. It was suggested to act as an inhibitor of the glucose transporters, GLUT2 and GLUT4, and thus interfere with the normal uptake of glucose [47]. Other mechanisms were proposed as well, such as agonistic activity towards GPCRs (G protein-coupled receptors) [48]. Myricetin affects cholesterol levels as well as being anti-inflammatory, and these functions may also afford some benefit in diabetes [46]. Myricetin has demonstrated antiviral and neuroprotective activity in vitro. It was also suggested to be of benefit in atherosclerosis and in vascular conditions through its antithrombotic activity, though little data are available to support this. Overall, clinical data proving that myricetin is medicinally useful are still missing. It is marketed as a dietary supplement and is also available in dihydro form (ampelopsin; **19**), Figure 3, which was also claimed to be of benefit in various diseases. It was especially promoted in alcohol use disorders, exerting its effect via interaction with GABA receptors [49].

Catechins (**12**–**15**), Figure 2, existing as mixtures of diastereomers (–)-catechin, (+)-catechin, (–)-epicatechin, and (+)-epicatechin, are quite abundant in food plants, though cocoa (*Theobroma cacao*; Malvaceae) seems to be an exceptionally rich source. The pharmacopeia of 100 years ago included catechin formulations for the treatment of various cardiovascular problems. Recent dietary studies showed that catechins might have a protective effect against cardiovascular diseases, such as hypertension [50,51]. Very interestingly, it was not long ago that the European Food Safety Authority (EFSA) panel stated that cocoa flavanols might be effective in maintaining endothelium-dependent vasodilation and normal blood flow [52]. Conversely, another recent study declared that there was no reliable association between catechin intake and the risk of cardiovascular disease [53]. Ingested catechins, as the majority of other flavonoids, seem to be rapidly metabolized and excreted from the body. Their mode of action has not been fully established, but they appear to affect platelet aggregation, are capable of decreasing blood pressure and glucose levels, and counteract dyslipidaemia [54,55]. It is questionable to what extent this activity can be ascribed solely to catechins, since catechin-rich foods such a cocoa contain a myriad of other compounds. Catechins might also produce adverse effects—the ingestion of some catechin-containing products was associated with haemolytic anaemia and renal failure triggered by an autoimmune reaction [56]. Some drugs containing catechins were withdrawn from the market after this discovery.

Anthocyanidins are unusual in that their oxygen at C_1_ is present in a positively charged state, commonly known as flavylium cation. Some examples of an anthocyanidin structure such as pelargonidin (16) and cyanidin (17) are shown in Figure 2. These compounds are commonly distributed in plants and are generally responsible for the purple, dark blue, violet, and black colouration of various plant parts such as blueberries and purple cabbage, and the colour changes depend on pH [57]. With regard to clinical usefulness, many of the properties of other flavonoids will be valid for anthocyanidins—low bioavailability, rapid metabolism and high excretion rate, together with beneficial effects in the treatment of cancer, diabetes, cardiovascular diseases (CVD), and neurodegenerative disorders. Some recent studies have disputed the claim that anthocyanidin consumption has antioxidant, anticancer, and anti-aging effects [58]. There have been no large clinical trials proving that anthocyanidins have any therapeutic benefits in human disease.

Some flavonoids exist as oligomers, usually as dimers or trimers, such as procyanidin C_1_ derived from three epicatechin units (**20**) in Figure 4. They may also be formed from structures other than the flavan-3-ols, such as the anthocyanins, which incorporate flavan-4-ols and flavan-3,4-diols. These small polymers are collectively known as condensed tannins. The biological activity of these compounds is not significantly increased in comparison to their monomeric derivatives, and they may be readily depolymerized to their precursor monomeric forms upon digestion. Condensed tannins affect the flavour of food and drink through their astringency, and some such as the procyanidins are used as dietary supplements. These compounds are usually extracted from bark of maritime pine (*Pinus pinaster*; Pinaceae) under the name Pycnogenol^®^ and contain up to 70% procyanidins. This formulation was claimed to be useful in the treatment of an array of human diseases, with special emphasis on venous insufficiency and the related vascular conditions. As with many other flavonoid-derived dietary supplements, the evidence is insufficient to support such claims [59].

The hydroxy- group at position C_3_ on the heterocyclic C-ring may be substituted by different structures. Epigallocatechin gallate (EGCG; **21**), Figure 4, is an ester of epigallocatechin and gallic acid. EGCG is found in high quantities in unfermented green tea (*Camellia sinensis*; Theaceae) and is considered the active ingredient for the health benefits of drinking green tea. EGCG was said to have anti-inflammatory, anticancer, cholesterol-lowering, and GABA antagonistic activity, and to be beneficial in preventing CVD, diabetes, and obesity [60,61,62], and EGCG is available as a dietary supplement. None of this was proved satisfactorily in clinical trials and there is still little evidence that EGCG provides these health benefits in humans. Some studies even showed that the excessive intake of EGCG is associated with hepatotoxicity and liver damage [63]. It is worthy of note that green tea catechins (the so called sinecatechins) were the first botanical product to be approved as a prescription drug by the FDA [64]. Additionally, methylated catechin (from benifuuki green tea) has been tested as a potential anti-allergic agent (e.g., it reduced the symptoms of hay fever) [65]. More clinical data are needed to make any reasonable conclusion about this activity.

If the tea leaves are allowed to ferment (oxidise), much of the colourless catechins, including EGCG, are enzymatically converted into theaflavins and thearubigins, which are responsible for the darker colour of black tea. Theaflavin (**22**), Figure 4, might also be present as theaflavin-3-gallate, theaflavin-3′-gallate, and theaflavin-3-3′-digallate. Since theaflavins are structurally related to catechins and EGCG, they are assumed to possess similar biological activities, and are thought to be especially of value in cancer prevention and CVD. Few clinical trials on theaflavin have been done. One study found an association between the consumption of theaflavin-enriched tea extracts and reduced levels of cholesterol [66], though this effect was not observed when purified theaflavins were administered [67]. Theaflavins, tea extracts, and tea also demonstrated promising results in psychophysiological stress [68] and the alteration of microvascular function [69], though more data are required. What was not addressed was the possibility that other compounds may significantly contribute to the therapeutic efficacy of tea, and the clinical usefulness of theaflavins is still far from conclusive. Theaflavins were also investigated as potential antivirals, in particular against HIV, and anticancer effects such as inhibiting cell growth, survival and metastasis [70]. As far as we know, these activities were only tested in vitro.

Many flavonoids are present in a glycosylated form, the sugar units being substituted at different hydroxy- groups, most commonly at positions C_3_ or C_7_. Rutin (**25**), Figure 5, from buckwheat (*Fagopyrum esculentum*; Polygonaceae) and rue (*Ruta graveolens*; Rutaceae) and hesperidin (**24**), Figure 5, from various *Citrus* sp. (Rutaceae) are marketed as dietary supplements under the name vitamin P. These products are claimed to be of value in the treatment of cardiovascular conditions related to endothelial dysfunction, post-thrombotic syndrome, and venous insufficiency by reducing the fragility of blood capillaries and increasing the flexibility of blood vessels. To date, there is still little evidence of their use being safe and effective, though rutin was found to moderately reduce oedema in lower extremities in patients suffering from chronic venous insufficiency (CVI) [71,72]. Hesperidin was also suggested to lower cholesterol levels and blood pressure, but again, detailed evidence is lacking [73]. Based on in vitro experiments, vitamin P was also reported to possess potent anticancer and anti-inflammatory properties. In spite of being glycosylated, a modification that is believed to increase bioavailability, both rutin and hesperidin have low absorption levels and are readily metabolized and excreted. These factors limit their therapeutic potential.

Daflon is a purified, micronized fraction of flavonoids, chiefly composed of diosmin (90%; **27**), Figure 5, and smaller amounts of other flavonoids such as hesperidin. Diosmin is commonly found in members of the genus *Citrus* (Rutaceae) but has also been isolated from non-rutaceous plants (such as *Teucrium gnaphalodes*; Lamiaceae). Daflon’s pharmacological profile is similar to that of the aforementioned glycosylated flavonoids. There is some evidence that daflon increases venous tone and resistance in small blood vessels, suggesting that it might be of value in the treatment of some cardiovascular conditions, such as CVI and post-thrombotic syndrome [71,74]. Daflon has a somewhat awkward position because, in some countries such as France and the Czech Republic, it is available as prescription drug, while in the U.S. and other countries it is treated as a dietary supplement and not approved as a prescription drug because of a lack of clinical evidence. Recently, it was also reported to be helpful in the treatment of haemorrhoids [75]. No adverse effects have been noted for daflon.

Neohesperidin (**24**), Figure 5, and naringin (**26**), Figure 5, are found in large amounts in *Citrus aurantium* and *C. paradisi* (Rutaceae), respectively. These glycosides are perceived by most persons as intensely bitter. An array of biological activities was suggested for neohesperidin and naringin, including anticancer, anti-inflammatory, neuroprotective, antidiabetic, and as a preventive in cardiovascular disorders, but sufficient clinical data are not available to support these claims. Naringin, naringenin and hesperidin were thought to be the ingredients of grapefruit juice responsible for the inhibition of human cytochrome P450 enzymes, such as CYP3A4 and CYP1A2 that are necessary for the metabolism of some drugs [76]. However, it was later found out that the juice also contained considerable quantities of furanocoumarins, which turned out to be strong cytochrome P450 inhibitors, and it is now thought that they, rather than the flavonoids, are the main cause of the drug dose-elevating effect of grapefruit juice. Recently, naringin and hesperidin were implicated as inhibitors of organic anion-transporting polypeptides (OATPs), through which they may significantly contribute to this drug effect as well [77].

Interestingly, if the glycosylated flavonoids neohesperidin and naringenin, are converted to dihydrochalcone (DHC) derivatives, neohesperidin-DHC and naringin-DHC; **28** and **29**, respectively), Figure 6, their flavour changes from bitter to extremely sweet, 300–1000 times as sweet as sucrose. Neohesperidin-DHC is used as a non-saccharide flavour enhancer, artificial sweetener, and a masking agent of some bitter-tasting components in both food and the pharmaceutical industry. It has a noticeable fruity and liquorice-like off note which, however, restricts its use to those products where this taste is not disturbing. Naringin-DHC is utilized as a sweetener as well, nevertheless, apparently not as often as neohesperidin-DHC. Some pharmacological activities have been suggested for these compounds: neuroprotective activity, hepatoprotective, anti-inflammatory, and apoptosis-inducing activity [78]. It was also proposed that neohesperidin-DHC might provide antidiabetic effects by lowering blood sugar through the inhibition of α-amylase [79]. Again, the evidence is too slim to make any reasonable conclusions about the clinical efficacy of DHC-derived sweeteners. Some animal studies have indicated adverse side-effects such as CNS depression/activation, muscle-relaxation/convulsion, or embryotoxicity [80]. No health complications have been observed with the regular consumption of neohesperidine-DHC.

Some of the information concerning the major flavonoids described above are summed-up in Table 3. This table also contains information on doses at which these compounds were tested for their biological activities (for animals, and where possible for humans), the observed toxic doses (for animals), and the daily doses recommended by the dietary supplement retailer. Note that the doses tested on animals are generally significantly higher than those tested on human subjects and those present in dietary supplements. It is particularly worthy of note that to date, the recommended daily dose has not been established for most (if not all) flavonoids.

## 3. Aromatic Polyketides with More Restricted Distribution

### 3.1. Synthetic Flavonoids

*Dysoxylum malabaricum* has a long tradition in Ayurveda as a medicinal plant used in the treatment of rheumatoid arthritis. Related species *D. gotadhora* (syn. *D. binectariferum*; Meliaceae) contains rohitukine, which was found to possess strong anti-inflammatory and immunomodulatory properties. It was initially thought that rohitukine is plant derived. However, it has now been shown that rohitukine is probably not biosynthesized by the plant itself but is the product of endophytic fungi. Rohitukine (**30**), Figure 7, belongs to the class of chromone alkaloids. While not being a flavonoid itself, structure–activity studies of rohitukine analogues led to the discovery of some synthetic compounds with structures similar to those of common flavonoids. Flavopiridol (alvocidib; **31**), Figure 7, exerts its anticancer properties by inhibiting various cyclin-dependent kinases (Cdk) enzymes, a relatively new target in anticancer therapy. It was successful in preclinical studies as well as phase I and II clinical trials as being effective against solid tumours, lymphomas and leukaemia [86]. Further studies have shown flavopiridol to have a rather narrow therapeutic index and relatively low level of selectivity, acting on non-Cdk targets. To overcome these limitations, a second generation of Cdk-inhibitors was developed. Most of these, however, no longer share the flavonoid skeleton. One such agent, dinaciclib, showed considerable promise in phase III trials, and this second generation Cdk-inhibitor seems to be far more effective than its flavonoid-like antecedent. Another flavonoid alkaloid with potent anticancer activity was P276-00 (riviciclib; **32**), Figure 7, which has so far made it to a phase II clinical trial [87].

### 3.2. Stilbenes

Perhaps the most extensively studied stilbene is resveratrol, existing as two cis-trans isomers (**33** and **34**), Figure 8. It is a constituent of red grapes and wines, and in recent decades, it has received considerable research attention as a molecule with alleged antioxidant and anti-inflammatory activity, the ability to inhibit platelet aggregation and to afford protection against cancer and cardiovascular problems. Additionally, targeting cancer stem cells (CSCs) in colorectal cancer (CRC) enhances CRC chemosensitivity to chemotherapeutic agents such as 5-fluorouracil, increases apoptosis and supresses migration and metastasis [88,89,90]. Dietary sources of resveratrol include wild blueberries, mulberries, and raspberries and peanuts. Commercial resveratrol is obtained from Japanese knotweed (*Reynoutria japonica*; Polygonaceae) and is sold as a dietary supplement. Despite its popularity with the public and the scientific community, as well as its reputation as a miracle molecule offering tremendous benefits to human health and well-being, it still has not been accepted as an effective therapeutic agent in medical practice in the U.S. Though showing promise in in vitro studies and animal models, resveratrol has failed to provide any significant therapeutic advantage over placebo in human clinical trials of cancer, cardiovascular disorders, metabolic diseases including diabetes and obesity, nor was it shown to have positive effects on longevity [91,92,93,94]. With regard to antioxidant activity, various studies have pointed out that resveratrol behaves as a strong pro-oxidant in vivo [23,95]. Other problems that hinder its clinical use is its poor water solubility and high rate of metabolism and excretion [96]. Because of the low bioavailability, some studies utilized high doses of resveratrol, which resulted in significant side-effects, including diarrhoea, nausea and weight loss [97,98]. To make things worse, resveratrol has been flagged as a pan-assay interference compound (PAIN) (compare with curcumin) that can produce false positive results in many different laboratory assays [99]. Analogues of resveratrol with better bioavailability are now in the process of development and clinical testing [100]. One current research focus for resveratrol is its ability to reduce secondary damage in stroke and traumatic CNS injury [101], to slow down cognitive impairment in Alzheimer’s disease [97], and preventive in skin cancer [102]. Nevertheless, the results are still preliminary.

Another stilbene of considerable medicinal interest is combretastatin A-4 (**35**), Figure 9, together with related structures (such as combretastatin A-1 and B-1) [104]. It is found in the bark of the bushwillow tree (*Combretum caffrum*; Combretaceae) native to the Cape region of South Africa. Combretastatins have a structure similar to that of resveratrol, however, they differ in substitution pattern and in that, the double bond has a *cis*-configuration, although combretastatins exist in a number of different isoforms. Combretastatins exert strong cytotoxic activity by binding to the colchicine site on tubulin and preventing its polymerization into microtubules. Combretastatin A-4 is structurally similar to colchicine, a strong antimitotic poison, and thus has attracted interest as a potential anticancer drug with similar mechanism of action as other commonly used antimitotic drugs such as analogues of vincristine [105,106], and it appears that they have selective toxicity toward tumour vasculature. Combretastatin A-4 provided the template for the synthesis of the water-soluble phosphate pro-drug combretastatin A-4 phosphate (CA4P; **36**), Figure 9. Based on its specific mode of action, CA4P stands as the lead drug of a novel class of depolymerising compounds known as vascular-targeting agents (VTAs). CA4P appears to specifically target established tumour blood vessels and rapidly reduce blood flow in the tumour, which results in potent anticancer activity [107,108,109]. This compound demonstrated promise in clinical trials as an anticancer agent against various solid tumours such as phase II/III ovarian cancer [110], but it is still in clinical development. Another combretastatin A-4 derivative with VTA-activity is ombrabulin. Nevertheless, it failed in the phase III clinical trial and the company that was developing this compound ceased further research [111]. Numerous analogues based on the combretastatin structure have been developed, and many of these have demonstrated significant anticancer activity in preclinical settings [86,112,113].

### 3.3. Styrylpyrones and Diarylheptanoids

As noted above, flavonoids and allied compounds are formed via a combination of *p*-coumaroyl-CoA and three units of malonyl-CoA. This is the most common biosynthetic pathway for aromatic polyketides. The *p*-coumaroyl-CoA starter unit may be extended with one or two malonyl units, giving rise to a subgroup of compounds with a considerably limited spread in the plant kingdom. In styrylpyrones, for example, two C_2_ units were added via malonate and the terminal chain often undergoes cyclisation to give lactones. Yangonin and the related kavalactones from kava (*Piper methysticum*; Piperaceae) provide one of the best illustrations of such compounds.

Aqueous extracts of the underground parts of kava have long been used in the South Pacific countries of Polynesia, Melanesia, and Micronesia as a drink with intoxicating properties. Kava has also gained popularity in herbal medicine for providing significant relief from anxiety, nervous tension, agitation, insomnia, headache and migraine [114]. There is mounting evidence that the pharmacological activity of kava stems from various kavalactones. To date, more than 18 kavalactones have been described, the predominant ones being the four enolides, kavain, methysticin, and their corresponding dihydro-derivatives, and two dienolides, yangonin and demethoxyyangonin (**37**–**42**), Figure 10. The anxiolytic properties of kava have been confirmed in clinical trials, though more evidence is needed to support its use in clinical practice [115,116]. Isolated kavalactones have also demonstrated analgesic, anticonvulsive, and relaxing activity on central muscles. Some of the kavalactones appear to affect neurotransmitter systems, including those where glutamate, γ-aminobutyric acid (GABA), dopamine, and serotonin are involved. Yangonin has also been shown to interact with cannabinoid receptors [117]. However, there have been reports of severe liver damage in patients consuming kava extracts on a regular basis. These cases of hepatotoxicity have raised questions of the safety of kava products. What was not sufficiently emphasized is that many of the kava users were taking extracts prepared using organic solvents such as acetone and alcohol. The hepatotoxicity might thus have been associated with the mode of preparation, rather than the kavalactones themselves [118]. Liver poisoning was also observed in people taking kava in a traditional aqueous form [119]. As a result, kava has been banned from sale in some countries. The problem was revisited and now it seems that cases of poisonings may have been related to low-grade kava material containing fungi with the possibility of hepatotoxic constituents such as aflatoxins and ochratotoxin A [120]. There is also an alternative explanation that some of the cases may have developed hepatotoxicity due to the specific reaction of their individual metabolic subtype [121]. At any rate, it should be noted that there are not enough data to prove that kavalactones cause liver damage. In addition to the hepatotoxicity problem, there is now ample evidence that kava can alter the metabolic breakdown of some prescription drugs, presumably by the inhibition of cytochrome P-450-dependent enzymes [122]. Kava is marketed as a dietary supplement, but it still remains peripheral to mainstream medicine and is not available as a pharmaceutical drug. Commonly used anxiolytics like benzodiazepines and barbiturates are strong enhancers of GABA receptors and they can cause serious adverse effects. Thus, kavalactones would provide a safer alternative in anxiety and the related disorders if their hepatotoxicity issue could be resolved. In addition to the promising results in anxiety, kava and kavalactones might also be of value to migraine sufferers, who have said that kava helped in reducing some of their symptoms, although there are no clinical data to back this up [114].

Another less well known subgroup of the aromatic polyketides that are attracting considerable research interest are the diarylheptanoids. They are formed by the condensation of coumaroyl-CoA and a single malonyl-CoA unit, with an additional cinnamoyl group added in the last step, hence the name diarylheptanoids. Curcumin (**43**), Figure 11, is the best studied member of this subgroup. It is found in the rhizomes of the turmeric plant, *Curcuma longa* (Zingiberaceae), where it is responsible for the typical yellow colour. It is used in the food industry as a colourant and assumed shelf life prolonger (under code E100). There is extensive evidence that curcumin has anti-inflammatory, anti-ulcer, anti-Alzheimer’s, anticancer, anti-arthritic and preventive properties against cancer, though none of its medicinal activities have been confirmed in clinical trials [123,124,125]. Its activity in vivo is limited by its instability, poor water solubility and low bioavailability, with nearly all the ingested amounts being rapidly excreted. Few authors discuss its non-selective pan-assay interference (PAIN) character, interacting with almost every pharmacological target and thus showing false positive results in many drug discovery assays [126,127]. Turmeric rhizomes contain other constituents besides curcumin that might be responsible for some of the therapeutic benefits of the plant in Ayurveda. Based on the evidence so far, it is very improbable that curcumin will provide an important lead for drug development. A few side effects such as nausea, diarrhoea and skin rash were observed [128]. Quite recently, curcumin is also available in the form of micronized powder. This formulation is considered to have increased bioavailability and therefore a higher rate of side effects can probably be expected [129]. Additionally, there have also been reports of deaths upon the intravenous application of curcumin as a part of naturopathic practice [130].

Some diarylheptanoids are alternatively formed via the incorporation of fatty acyl thioester (rather than another coumaroyl-CoA group) to the terminal chain of the cinnamoyl-CoA. Resulting compounds include gingerols and shogaols found in the rhizomes of ginger (*Zingiber officinale*; Zingiberaceae) that are responsible for its pungency. The chain lengths in the fatty acyl group may differ, giving rise to variants analogous to the parent compounds—e.g., 6-, 8-, or 10-gingerol and 6-, 8-, 10-shogaol (utilizing six-, eight-, or ten-carbon fatty acid fragments, respectively; **44**–**49**), Figure 12. Ginger has been utilized for centuries as a culinary and medicinal plant, being traditionally used in the treatment of a wide range of human diseases, including colds, pulmonary disorders, rheumatic pain, headache, gastrointestinal problems, and motion sickness (kinetosis). Its medicinal properties have been, to some degree, confirmed in clinical trials such as in the case of migraine. Again, various studies assume that gingerols and shogaols are responsible for much of the therapeutic activity of ginger, though the clinical trials used ginger extracts, not the pure compounds. Gingerol-standardized extract of ginger demonstrated anti-inflammatory activity in an animal model of rheumatoid arthritis, though non-standardized extract showed efficacy as well [131]. Shogaol was shown to inhibit one of the serotonin receptors, 5-HT_3_, which appears to be associated with the anti-nausea (anti-kinetosis) activity of ginger [132]. Recently, 6-gingerol and 6-shogaol showed pain-relieving activity via the inhibition of voltage-activated Na^+^ currents in an in vivo model [133]. Ginger also contains other constituents, including terpenoids, such as the sesquiterpene zerumbone, that have also displayed significant anti-inflammatory and agonistic activity toward 5-HT_1A/B_) [132,134]. This suggests that gingerols and shogaols are not the only active principles of ginger rhizome.

### 3.4. Flavonolignans

Flavonolignans are an interesting class of compounds that arise from a combination of flavonoid and phenylpropanoid (usually coniferyl alcohol) structures via oxidative coupling between the two. Perhaps the best example of such compounds that have attracted significant research and medicinal interest are flavonolignans from milk thistle (*Silybum marianum*; Asteraceae). It is a biennial plant found in the Euro-Mediterranean region, where it has a long usage in traditional medicine being used as a remedy for liver problems and other disorders. The seeds contain 1.5–3% of flavonolignans, collectively termed silymarin, which is composed of silybin (also known as silibinin), together with silychristin, isosilybin, and related structures (**50**–**54**), Figure 13. Silybin and isosilybin are equimolar mixtures of two trans-diastereomers. Silymarin is currently available in various preparations of milk thistle extract, which are typically composed of about 80% flavonolignans [135,136,137]. Various in vitro experiments and in vivo animal studies have demonstrated that silymarin has the potential to treat liver disease and injury, including hepatitis, cirrhosis, and jaundice. However, the results from early clinical trials returned conflicting data [138]. Silybin, which appears to be chiefly responsible for the pharmacological activity of silymarin, has poor water solubility and low bioavailability. Various silybin analogues with better solubility properties have been developed and are currently being tested in clinical trials for treating hepatic damage. Perhaps the most successful outcome of such effort is silybin bis-hemisuccinate (e.g., Legalon^®^ SIL), which has been shown to be an antidote in acute poisoning with α-amanitin and phalloidin after the ingestion of death cap mushrooms (*Amanita phalloides*; Amanitaceae) and is now used for this purpose in medicinal practice [139]. The administration of silybin-phosphatidylcholine resulted in improvement in patients with non-alcoholic fatty liver disease in a phase III clinical trial [140]. One problem with current silybin derivatives for wider clinical use was the fact that they cannot be taken orally but must be injected [138]. More work needs to be done in this field.

### 3.5. Isoflavonoids

Isoflavonoids comprise a specific subgroup of structural variants of flavonoids, where the aromatic B ring has migrated to the neighbouring carbon on the heterocyclic C ring (from position C_2_ to position C_3_), which requires the activity of isoflavone synthase (IFS) enzymes. Initially it was believed that only the leguminous plants do possess the necessary biosynthetic apparatus (i.e., IFS enzymes) for the biosynthesis of isoflavonoids, and thus that these compounds are entirely limited to members of the Fabaceae family. However, isoflavonoids have now been detected in more than 60 non-leguminous taxa, including those with medicinal importance, such as Apocynaceae, Amaryllidaceae, Iridaceae. The distribution of Malvaceae, Poaceae, Rubiaceae, and Zingiberaceae [141,142], however, remains restricted in comparison to common flavonoid structures. As with flavonoids, isoflavonoids are prone to further modifications, including hydroxylation and alkylation, various levels of oxidation of the heterocyclic ring, or the formation of additional heterocyclic rings, which create a high level of structural complexity. Many natural derivatives of the basic isoflavone skeleton have been described, such as isoflavans, prenylated isoflavonoids, coumestans, pterocarpans, and rotenoids). It appears that plants produce isoflavonoids for defensive purposes, especially against fungi [4,143].

Isoflavonoids have attracted a lot of research attention because of their potent phyto-oestrogenic effects—particularly those of simple isoflavones like genistein and coumestans such as coumestrol. One of the important sources of these compounds are common grazing plants (such as *Trifolium* spp.; Fabaceae). The phyto-oestrogenic effect was first observed in animals that were grazing on these fabaceous plants as their reproductive ability was significantly altered. Isoflavonoids exert oestrogenic activity by mimicking the structure of the steroid hormone oestradiol and binding to the corresponding receptors. Their relative activity is much less than that of oestradiol. Isoflavonoids were of interest as agents combating the negative symptoms of menopause, such as hot flushes, mood swings, sleep disturbance, tiredness, osteoporosis, and cardiovascular problems. These symptoms are currently treated by the administration of oestrogens as hormone replacement therapy. Though efficient in reducing menopausal problems, the use of oestrogens was associated with thromboembolic events and an increased risk of breast and ovarian cancer. It was thought that isoflavonoids might provide a safe alternative for the relief of menopause symptoms, and it was also claimed that they had other beneficial effects including protection against various types of cancer and Alzheimer’s disease. The clinical efficacy of isoflavonoids is still controversial and available data do not support any unequivocal conclusions [144]. Positive effects were observed in only a few studies, such as in osteoporosis [145]. Others such as the prevention of hot flashes failed to show any benefit [146], while in the case of cardiovascular problems and cancer prevention, the results were inconsistent [147,148,149,150]. Since isoflavonoids both stimulate and antagonise the oestrogenic response, this potential for endocrine disruption has sparked debate and health concerns about the risks associated with dietary exposure to isoflavonoids. It was claimed that the frequent consumption of isoflavonoid-containing foods such as soy products might alter the normal levels of testosterone in males and cause infertility, but to date there has been no evidence to prove this claim [151,152,153]. Soy products are widely used in infant formulas, and questions were raised about whether phyto-oestrogens influenced the normal development, growth, and reproduction when consumed at an early age. Again, available studies have not shown these functions to be adversely affected [154,155].

The aforementioned information raises doubts about the safety and usefulness of isoflavonoids in menopause and other diseases. After years of testing alternatives, hormone replacement therapy (HRT) still seems to provide the best therapy, at least for osteoporosis. Isoflavonoids have not been accepted by conventional medical practice, but they are available as dietary supplements primarily from soybean (*Glycine max*; Fabaceae), which contain high levels of daidzein and genistein in free or 7-*O*-glucosidic forms and red clover (*Trifolium pratense*; Fabaceae), whose isoflavones are chiefly composed of formononetin and daidzein free or as 7-*O*-glucosides; **55**–**58**), Figure 14. Another compound that has been largely discussed as a potential phytoestrogen is miroestrol (**65**), Figure 15, from *Pueraria mirifica* (Fabaceae). Miroestrol is available in the form of standardized extracts of *P. mirifica* and is marketed as dietary supplement to treat the symptoms of menopause, similar to other more common isoflavonoids (daidzein, genistein). However, much of the miroestrol claims have not been sufficiently proved in clinical trials, and the Federal Trade Commission has taken legal action against some marketers of these products. Other isoflavonoids, such as vestitol (**59**), medicacarpin (**61**), and pisatin (**62**), Figure 14, seem to be only active in plants for defence against fungal attack, and are not of interest as phytoestrogens.

A few analogues of isoflavonoids have been developed as drugs. Phenoxodiol (idronoxil; **66**), Figure 16, a simple isoflavone, showed promise in clinical trials as an anticancer agent with an uncommon mode of action among naturally derived anti-tumour drugs. It inhibits NADH oxidase (tNOX) in cancer cells, which triggers apoptosis. Phenoxodiol is still being tested in late-phase clinical trials against a wide range of cancer types, including chemo-resistant ovarian cancers as well as prostate and cervical cancers [156,157]. Some companies have discontinued phenoxodiol research because of the development of analogues with better properties, such as ME-344 (**67**) Figure 16 [158]. Some of the more complex isoflavonoids were also found to be medicinally useful. Rotenone and deguelin (**63** and **64**, respectively) Figure 14 are found in roots of various *Derris* (e.g., *D. elliptica* and *D. malaccensis*; Fabaceae) and *Deguelia* sp. (e.g., *D. utilis*, syn. *Lonchocarpus utilis*; and *D. rufescens* var. urucu, syn. *L. urucu*; Fabaceae).

The dried, powdered roots have long been used as insecticides. The insecticidal action of rotenone and deguelin is achieved via their ability to complex with the NADH: ubiquinone oxidoreductase of the respiratory chain, blocking electron transport to ubiquinone and disrupting oxidative phosphorylation. Insects and fish lack the enzymes for rapid detoxification, while mammals are not susceptible to rotenoids because they are able to rapidly metabolize them when ingested. Rotenoids pose a serious threat only when they enter the blood stream. Rotenoid-producing plants were traditionally used for “lazy fishing” in which the powdered root was sprinkled in water, killing all nearby fish. When the fish were eaten, the diners suffered no adverse effects. With their non-toxic profile and biodegradability, rotenoids are still frequently used as insecticides. Rotenone is used in human medicine to treat head lice and scabies and in veterinary practice in the management of mites, lice and ticks [4]. It is however being superseded by agents with improved stability and insecticidal activity, such as semi-synthetic derivatives of pyrethrin. Rotenone is also utilized in rat models of Parkinson’s disease, although the validity of this model has been questioned [159]. In addition, rotenone has found considerable use as a piscicide for the elimination of non-native fish species. The aforementioned data are summed-up in Table 4; this table also provides the recommended areas of treatment, the proposed mode of action, and the dosage and the routes of administration for most of the discussed compounds.

## 4. Conclusions

Natural products have always attracted interest as leads in the search for new pharmaceutical drugs. Among all the plant-derived natural compounds, flavonoids and their structurally related compounds have received a great deal of attention over the last decade. Flavonoids and plant-derived aromatic polyketides have been claimed to be beneficial in a vast array of human diseases from cancer and diabetes to cardiovascular problems. The outcome of all the effort that has been invested in researching flavonoids as possible drugs may be perceived as rather disappointing since few have actually proceeded to clinical use. Some of the major problems hindering their clinical use are low water solubility, rapid metabolism and excretion, and weak efficacy. Perhaps the only flavonoid-derived compound that has been approved for medicinal use is the injectable form of silymarin used in the treatment of acute poisoning with the mushroom, Amanita phalloides. Combretastatin A-4 phosphate, an agent with a novel mode of anticancer action, is in the later stages of clinical development. Phenoxodiol and the related analogues also demonstrated promising anticancer effects in clinical trials. Other flavonoid-derived analogues, despite showing considerable promise, appear to have some drawbacks, and in some cases, have been superseded by more effective agents such as analogues of rohitukine, or like the kavalactones, have severe adverse effects, or are orally inactive and have poor bioavailability like silymarin. Several studies suggest that promising dietary flavonoids and allied compounds such as quercetin, resveratrol, and curcumin have not shown consistently significant clinical efficacy in humans (to date). Thus, do the flavonoids and related plant-derived aromatic polyketides deserve to be called miracle compounds? Some of them such as combretastatins, phenoxodiol-related compounds, silymarin, and kavalactones, perhaps yes. The clinical efficiency of the other members is at least questionable. Only the course of time will tell.

## Figures and Tables

**Figure 1 molecules-25-03846-f001:**
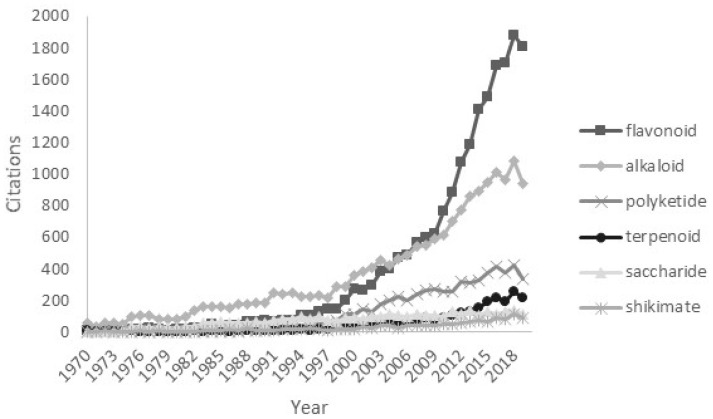
The number of research articles containing the names of the main groups/subgroups of natural products in the title/abstract from 1967 to 2019. Data obtained from PubMed on 22 September 2019.

**Figure 2 molecules-25-03846-f002:**
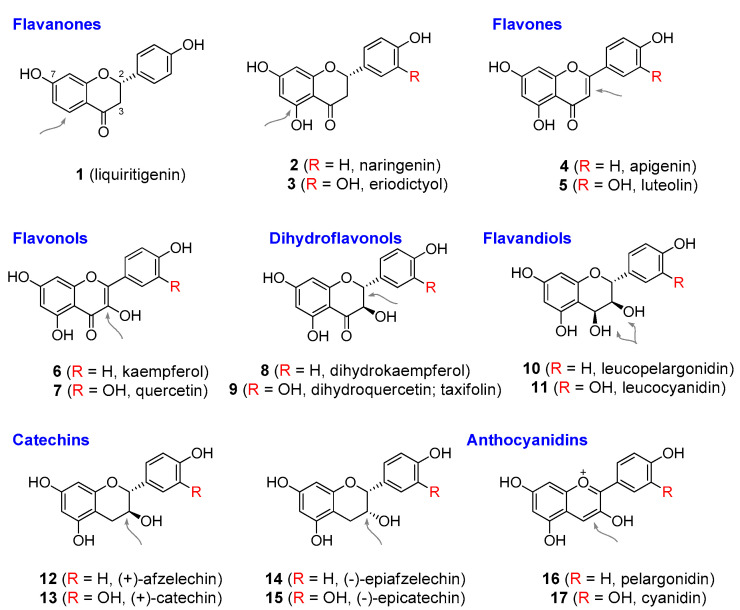
Molecular structures of the common flavonoids.

**Figure 3 molecules-25-03846-f003:**
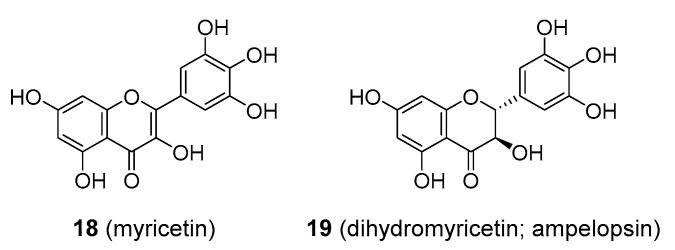
Molecular structures of myricetin and its analogue.

**Figure 4 molecules-25-03846-f004:**
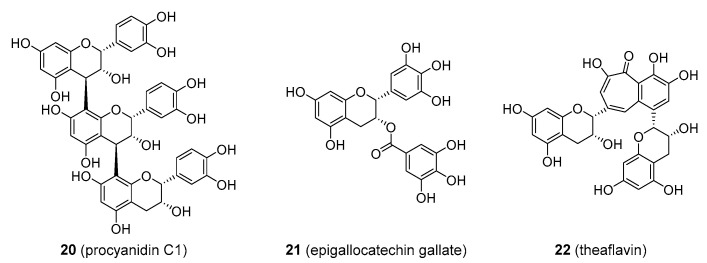
Flavonoids with higher levels of hydroxylation/complexity.

**Figure 5 molecules-25-03846-f005:**
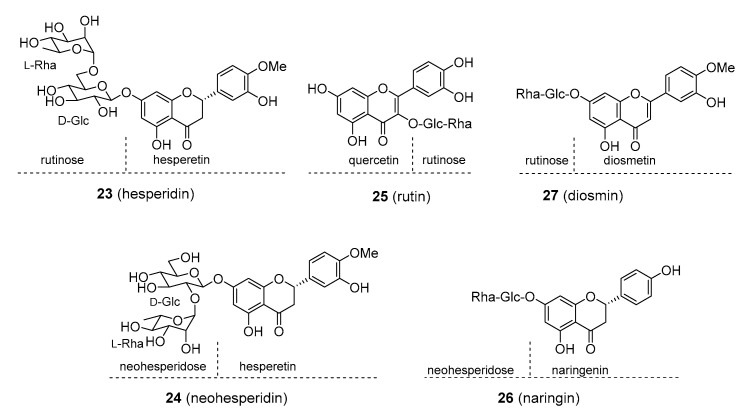
Molecular structures of some of the glycosylated flavonoids.

**Figure 6 molecules-25-03846-f006:**
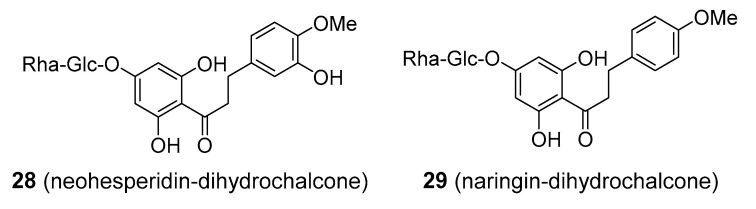
Glycosylated dihydrochalcones used as non-sugar sweeteners.

**Figure 7 molecules-25-03846-f007:**
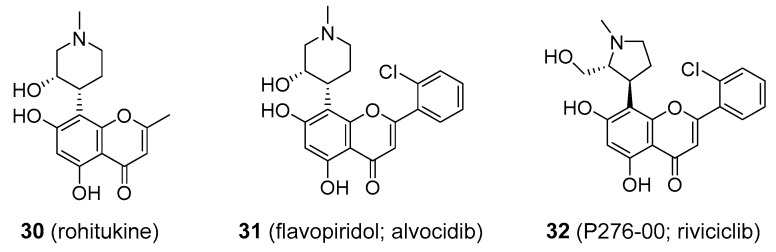
Natural rohitukine and some of its synthetic derivatives with anticancer properties.

**Figure 8 molecules-25-03846-f008:**
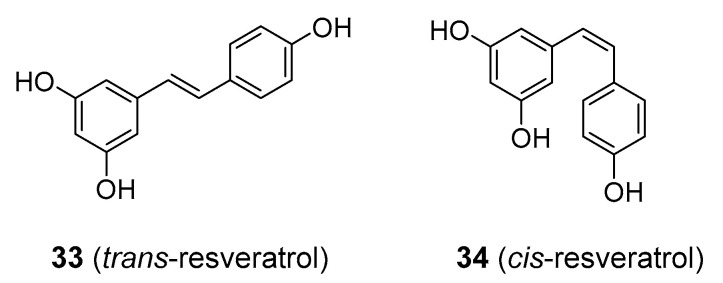
Resveratrol exists as cis- and trans-isomers, the trans- form being prevalent in nature. No significant difference in biological activity was observed between the two [103].

**Figure 9 molecules-25-03846-f009:**
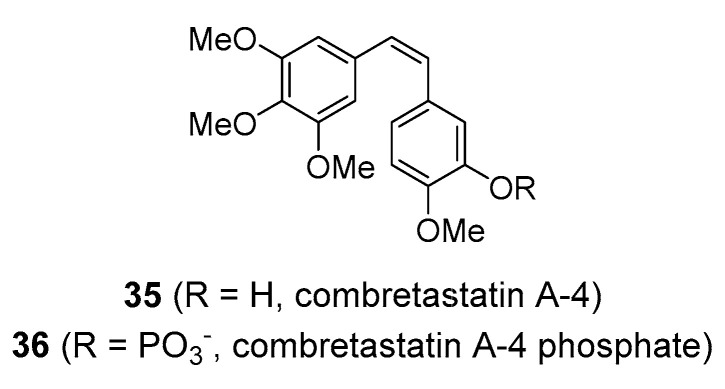
Molecular structures of combretastatins.

**Figure 10 molecules-25-03846-f010:**
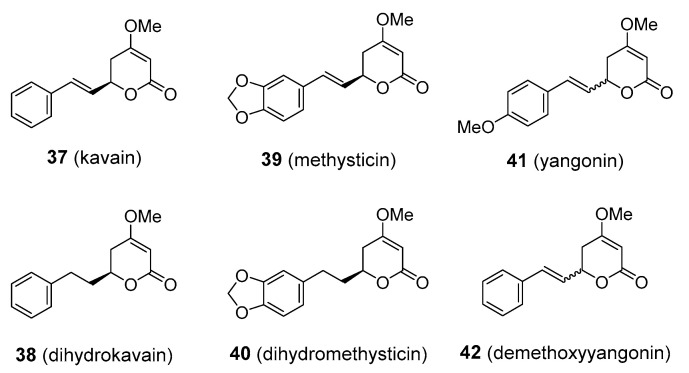
Molecular structures of kavalactones.

**Figure 11 molecules-25-03846-f011:**
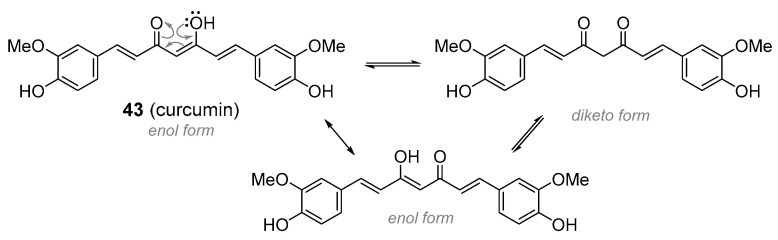
Curcumin exists as a tautomeric mixture of ketoenol- and diketo-form. The ketoenol structure (depicted) is more common in nature.

**Figure 12 molecules-25-03846-f012:**
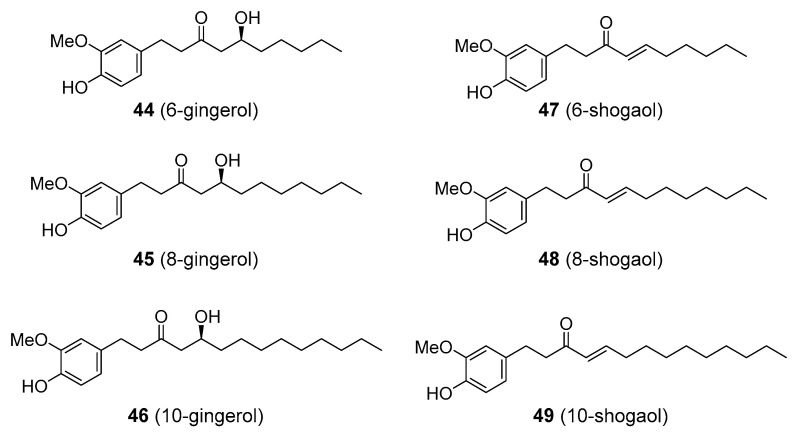
Compounds related to the diarylheptanoid pathway.

**Figure 13 molecules-25-03846-f013:**
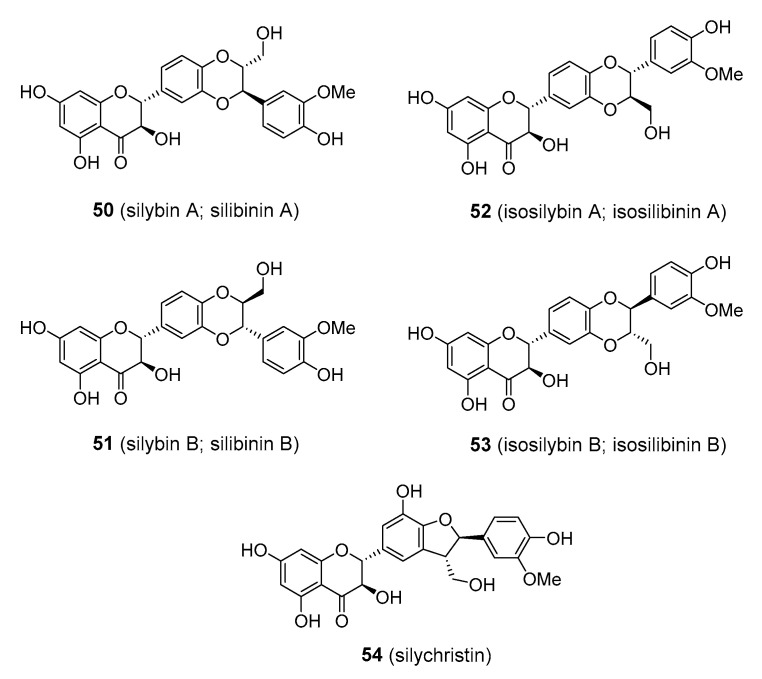
Flavonolignans from the milk thistle (*Silybum marianum*; Asteraceae) collectively termed as silymarins.

**Figure 14 molecules-25-03846-f014:**
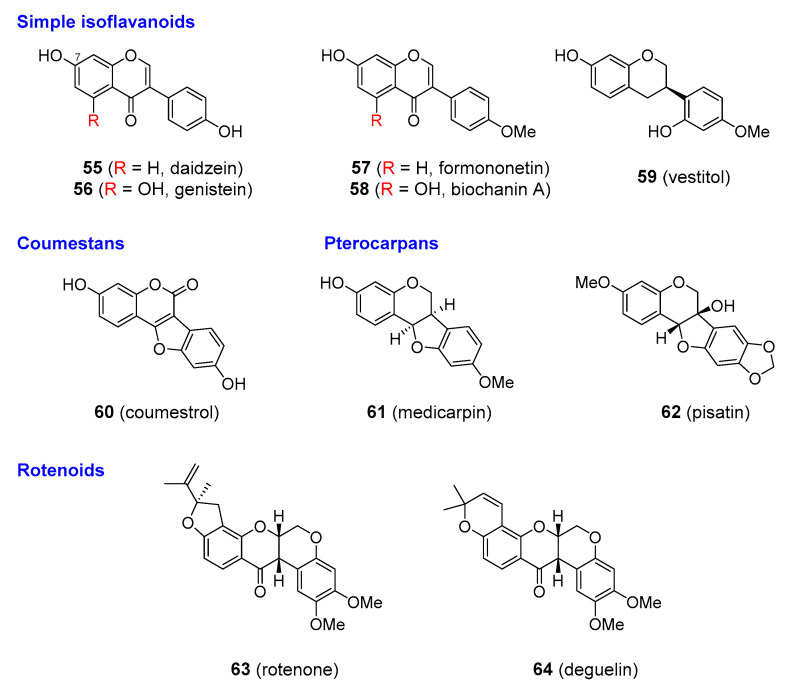
Isoflavonoids of common and more restricted distribution.

**Figure 15 molecules-25-03846-f015:**
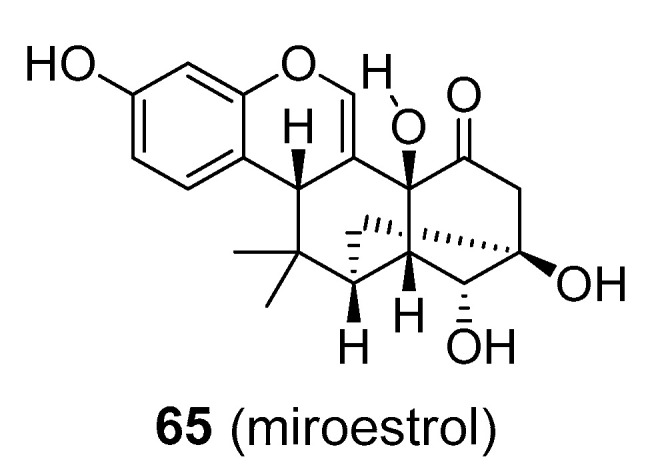
Molecular structure of miroestrol form *Pueraria mirifica* (Fabaceae).

**Figure 16 molecules-25-03846-f016:**
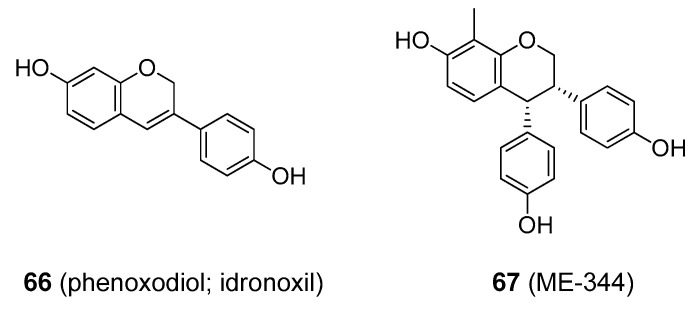
Some of the simple isoflavan derivatives that showed promising anticancer activity.

**Table 1 molecules-25-03846-t001:** Common sources of dietary flavonoids.

Compound	Common Sources
apigenin	Vegetables of the Apiaceae family, such as parsley and celery
luteolin	
kaempferol	Fruits (apples, cherries, berries), brassicaceous vegetables (broccoli, Brussels sprouts, cabbage), amaryllidaceous plants (onions, leeks), beverages (tea, red wine)
quercetin	
myricetin	
rutin	
catechin	Green tea, cocoa, chocolate, alcoholic beverages (red wine), some fruits (apples)
epicatechin	
epigallocatechin gallate	
theaflavin	Black tea
cyanidin	Fruits and beverages (berries, cherries, grapes, red wine)
pelargonidin	
hesperidin	Citrus fruits (lemons, oranges, grapefruits), grapes and some vegetables (e.g., tomatoes)
neohesperidin	
naringenin	
naringin	
taxifolin	

**Table 2 molecules-25-03846-t002:** Sources of aromatic polyketides with more restricted distribution.

Group/Compound	Source
**Stilbenes**	
resveratrol	Grapes (*Vitis vinifera*; Vitaceae), cherries (various *Prunus* species; Rosaceae), groundnuts (*Arachis hypogaea*; Fabaceae), Japanese knotweed (*Reynoutria japonica*; Polygonaceae)
combretastatin A-4	Eastern Cape South African bushwillow tree (*Combretum caffrum*; Combretaceae)
**Styrylpyrones**	
kavalactones	Kava kava (*Piper methysticum*; Piperaceae)
Diarylheptanoids	
curcumin	Turmeric (*Curcuma longa*; Zingiberaceae)
gingerols	Ginger (*Zingiber officinale*; Zingiberaceae)
shogaols	
**Flavonolignans**	
Silymarin	Milk thistle (*Silybum marianum*; Asteraceae)
**Isoflavonoids**	
daidzein	Leguminous plants (such as soybean, *Glycine max*; Fabaceae)
genistein	
coumestrol	Lucerne and clovers (*Medicago sativa* and *Trifolium* spp; Fabaceae)
medicarpin	Lucerne
vestitol	
pisatin	Pea (*Pisum sativum*; Fabaceae)
rotenoids (e.g., rotenone, degueline)	Various *Derris* and *Deguelia* species (Fabaceae)

**Table 3 molecules-25-03846-t003:** How important are flavonoid derivatives in human disease?

Compound	Condition at Which It Might Be Particularly Helpful ^a^	Doses at Which It Was Tested (mg/kg Body Weight)/Type of Study	Doses at Which It Showed Toxic Effects (mg/kg Body Weight; Oral Doses in Animals) ^b^	Daily Doses Recommended by the Dietary Supplement Retailer (mg/kg Body Weight) ^c^	Is There Clinical Evidence That It Has, or Will Have, Therapeutic Benefit in Humans?
quercetin	cancer	1.6–4000/animal studies up to 14.3 ^c^/human studies	159	0.3–7.1	Some ^i^ [27]
kaempferol	cancer	1–200/animal studies	1000	1.4–5.7	Very limited [81]
taxifolin	cancer	~ 50 mg/animal studies	985–1200 IP ^d^	0.14–0.2	No
naringenin/ naringin	CVD	5–200 mg/animal studies 2.9 ^c/^human studies		0.2	Very limited [82]
apigenin	AD, cancer	7.5–50/animal studies	Data not available	0.7	No good data [43]
luteolin	cancer, inflammatory conditions	10–100/animal studies	>2500–5000	1.4–4.3	No [83]
myricetin	inflammatory conditions, diabetes	50–500/animal studies	1000 IP ^d^	1.4	Limited for humans [46]
catechin ^e^	CVD	50–2000/animal studies 1.4–7.1 ^c/^human studies	>10,000	0.7–8.6	Some ^i^ [84]
epicatechin ^e^			1000		
epigallocatechin gallate ^e^			2170		
theaflavin ^f^	CVD and cancer	250–3000/animal studies 1.4–7.1 ^c^/human studies	562 IP ^d^	0.7–1.4	Limited [66]
anthocyanins	diabetes	10–2000/animal studies 0.3–16.4 ^c^/human studies	Data not available	0.02–1.4	Limited [85]
pycnogenol^® g^	venous insufficiency	10–40/animal studies 2.1–5.1 ^c/^human studies	2000–4000	0.3–1.4	Limited [59]
rutin	venous insufficiency	10–150/animal studies 7.1 ^c^/human studies	2000 IP^d^	usually 7.1	Some ^i^ [71]
daflon ^h^	venous insufficiency	7.1–14.2 ^c^/human studies	>10,000 for diosmetin 1000 IP ^d^ for hesperidin	7.1–14.2	Some ^i^ [74]

^a^ AD = Alzheimer’s disease, CVD = cardiovascular disorders; ^b^ data retrieved from PubChem database (https://pubchem.ncbi.nlm.nih.gov/; accessed on 30 January 2020); ^c^ given that the average weight of an adult human is 70 kg; ^d^ intraperitoneal injection, oral data not available; ^e^ usually studied and marketed as green tea catechins (GTC) or standardized green tea extracts; ^f^ usually studied and marketed in the form of standardized black tea extract; ^g^ chiefly composed of procyanidins; ^h^ composed of diosmetin (90%) and hesperidin (~10%); ^i^ more data are needed to make any reasonable conclusions.

**Table 4 molecules-25-03846-t004:** The roles of aromatic polyketides with limited distribution in human disease.

Compound	Medicinal Application ^a^	Mode of Action	Dose and Mode of Application ^a^	Is There Clinical Evidence That It Has, or Will Have, Therapeutic Benefit in Humans? ^a^	Comment ^a^
flavopiridol	cancer	cyclin-dependent kinase inhibitor	60–100 mg mg/m^2^ IV	Yes [160]	Replaced by more efficient agents
resveratrol	inflammatory conditions, CVD, cancer	remains to be established	1.4–4.2 mg/kg bw ^b^ orally in the form of dietary supplements	Not enough data [92,100]	Still not accepted as a medicinal agent
combretastatin A-4 posphate	cancer	vascular disrupting agent; mitotic poison	5–120 mg/m^2^ IV	Yes [161]	Still in clinical development
kava and kavalactones	anxiolytic	interaction with GABA, glutamate, dopamine, serotonin and cannabinoid systems	1–3.5 mg/kg bw ^b^ of kavalactones orally in the form of standardized kava root extract	Yes [116]	Due to severe hepatotoxicity no longer marketed in many countries
curcumin	inflammatory conditions, cancer	remains to be established	7.14–14.2 mg/kg bw ^b^ of turmeric extract orally in the form of dietary supplements	Despite large number of clinical trials, no evidence have been observed as of yet [127]	There are few reports of deaths after administration of IV curcumin
gingerols and shogaols	kinetosis, migraine, headache, rheumatism	presumably via interaction with serotonin receptors	3.5–14.2 mg/kg bw ^b^ of ginger powder orally	Limited data in humans [132]	Ginger contains other compounds which may also contribute to observed effects
silymarin	liver damage and injury	inhibition of toxin absorption	2–10.2 mg/kg bw ^b^ of silymarin orally in the form of standardized *Silybum* seed extract 20 mg/kg bw ^b^ IV ^c^	Yes, e.g., IV form of silymarin^c^ is used clinically in treatment of mushroom poisoning. Oral products (e.g *Silybum* infusions) seems to have low effectivity [138]	Orally active derivatives would perhaps expand the therapeutic applicability
daidzein/genistein	menopause symptoms	interaction with oestrogen receptors	0.05 - 0.7 mg/kg bw ^b^ of isoflavonoids orally in the form of standardized soy extracts	Conflicting results are observed, more studies are required [144]	Oestrogens are planar molecules, whereas isoflavonoids not–this feature may impede their interaction with oestrogenic receptors
phenoxodiol	cancer	inhibition of NADH oxidase	up to 27 mg/kg bw ^b^ IV	Yes [162]	More efficient agents are in clinical development

^a^ CVD = cardiovascular disorders, IV = intravenous application; ^b^ given that the average body weight of an adult human is 70 kg; ^c^ silymarin in the form of bis-hemisuccinate (Legalon^®^SIL).

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
