# Peer review of "Flavonoids and Related Members of the Aromatic Polyketide Group in Human Health and Disease: Do They Really Work?"

_molecules, 2020, doi:10.3390/molecules25173846_

Round 1

Reviewer 1 Report

This manuscript describes functionality on health caused by flavonoids (in broader sense). The English seems to be OK, or even much better than of the referee, as a reading material, but the referee is afraid that it is insufficient to express chemical reactions in biosynthetic pathways or exact structures of compounds. As for pharmacological topics the authors describes sound natural much natural though.

The referee can not understand why the authors want to add biosynthetic pathway of flavonoids or its subclasses, or roles in plant. Biosynthetic pathway does not contribute the function of flavonoids in human health. 

Sorry the referee catches that the authors focused strongly on clinical evidence on functions of flavonoids.

L56 It is interesting to see how research attention may dramatically shift from one theme towards another.

This sentense is difficult to understand.

L58 ....has skyrocketed over

The vocabulary is inappropriate

 L66~68 Dewicks text book ,

itis almost just quotes, it is better to delete!

L109 2. Flavanoids and stilbenes The referee feels this section is unnecessary

L128 Flavonoids are not contained #exclusively in vascular plant # but some algae forms, if consider the structure of flavonoids, some marine bacteria produce the same skeleton by not shikimate pathway. Very old work by Vining!

L175 2.2.2. Health Benefits of Common Dietary Flavonoids and Stilbenes via Non-Antioxidant Mechanisms

The referee feels this section should written much compacted, if the conclusion will not be changed!

L232~L234 What is miracule? Because this is scientific review, proncipally the authors depict what is miraculous and negate each ones!

L279~L283 is well done!

L295~ EGCG should be describes at this position not later!

As for EGCG and catechins, "the only mixture of natural products" which was proved by FDA! The authiors should mention at least, or if the authors know about that, to negate it after depicting.

Shaw T Chen et al

NATURE BIOTECHNOLOGY VOLUME 26 NUMBER 10 OCTOBER 2008

L337 Cameria sinensis should be italic 

All plant names are written in roman, theses should be all changes!

L436, L461, L483....

L485 ..iin that the double bond has a Z-configuration, Double bond "does not has a Z-configuration" 

It is easy to read, I admit, but L484~L485 is so to apeak wordy!

L506 ....the most common synthetic pathway for

Biosynthetic

The refree does not recommend to use "may be" here, for bisnoryangonin derivaties are known to be biosnthesized by Type III

L568 The terminal chain of condensed cinnamoyl-CoA may incorporate a fatty acyl thioester rather

After curcumin topics, L568 is better start by " The terminal chains in plant heptanoids" to explain the topics changed to heptanoids...

L570 Zingiber officinale; italic

The referee considers that shogaol and sanshool are both required to prevent ireus especially after operations on colon cancer. I suppose it is established!

Hydroxy-_ sanshool induces colonic motor activity in rat proximal colon: a possible involvement of KCNK9 Am J Physiol Gastrointest Liver Physiol 308: G579–G590, 2015.

The below sentence is inappropriate!

L587 suggests that gingerols and shogaols are not the only active principles of ginger rhizome. Are they also pan-assay interfering compounds?

L596 italic

L624 believed that isoflavonoids were only found in plants of the Fabaceae family because they were the only plants with the necessary biosynthetic apparatus.

What apparatus means?

L681~L683 Names should be in italic.

L642~L646 Once the authors admitted the role of isoflavonoids in controlling

menopausal problems, at last the conclusion sounds it  should not be utilized. The referee feeles that the reader could be confused.    The function of 3"- methylated catechin in hay feber was established. If the authors discuss on curcumin, the micronized formulations curcumins should be handled. because micronization is recently often used, and sometimes it resulted very serious decease by its huge extension of bioavailability of compounds on target. As for esrogenic or non estrogenic activities of isoflavonoids, the referee feels why the authors did not pick up miroesterol in Pueraria mirifica?  

Author Response

ITEMIZED RESPONSE TO THE REVIEWER’S COMMENTS

Ms. Ref. No.: molecules-884414

Authors: Jan Tauchen, Lukáš Huml, Silvie Rimpelová, Michal Jurášek

Title: Flavonoids and related members of the aromatic polyketide group in human health and disease: Do they really work?

Reviewer 1:

Comments and Suggestions for Authors

Query No. 1: This manuscript describes functionality on health caused by flavonoids (in broader sense). The English seems to be OK, or even much better than of the referee, as a reading material, but the referee is afraid that it is insufficient to express chemical reactions in biosynthetic pathways or exact structures of compounds. As for pharmacological topics the authors describes sound natural much natural though.

Response: With all due respect, the authors do not understand the ground of criticism. The manuscript describes the clinical significance of flavonoid-derived substances; and for each substance, its structure is given (as is usually done in articles concerned with medicinal chemistry). These structures are drawn according to the rules of American Chemical Society (ACS).

Query No. 2: The referee can not understand why the authors want to add biosynthetic pathway of flavonoids or its subclasses, or roles in plant. Biosynthetic pathway does not contribute the function of flavonoids in human health.

Sorry the referee catches that the authors focused strongly on clinical evidence on functions of flavonoids

Response: With all due respect, the authors do not understand the ground of criticism. Yes, it is true that knowledge on a biosynthetic pathway of flavonoids does not contribute to their function on human health, however, the biosynthetic approach (and nomenclature of flavonoids and related compounds) yielding natural products described seems as a logical introduction to the paper. This section serves as a definition of terms and settings of barriers on substances which the article deals with further in the text. Not all readers are aware of the metabolic pathways leading to flavonoid biosynthesis, and nomenclature of these substances is sometimes misleading. We are convinced that this section has a key role in the manuscript.

Query No. 3: L56 It is interesting to see how research attention may dramatically shift from one theme towards another. This sentense is difficult to understand.

Response: We agree with the referee, this sentence has been removed from the manuscript.

Query No. 4: L58 ....has skyrocketed over. The vocabulary is inappropriate.

Response: We agree, the sentence has been changed for “has dramatically increased over the last decade”.

Query No. 5: L66~68 Dewicks text book , itis almost just quotes, it is better to delete!

Response: We agree with the referee, the sentence has been deleted.

Query No. 6: L109 2. Flavanoids and stilbenes The referee feels this section is unnecessary

Response: In this section, we wanted to demonstrate that flavonoids and related substances have an irreplaceable role in plants and are very important to them (indeed, they produce them for this purpose). However, this does not necessarily mean that they have a comparable importance (role) in humans. With all due respect, we are strongly convinced that this section has an important role in the manuscript (to the reader).

Query No. 7: L128 Flavonoids are not contained #exclusively in vascular plant # but some algae forms, if consider the structure of flavonoids, some marine bacteria produce the same skeleton by not shikimate pathway. Very old work by Vining!

Response: We would like to thank the referee for this suggestion. The sentence has been revised and changed: “Stilbenes and flavonoids are found mainly in vascular plants, primarily in angiosperms. However, some flavonoid structures have been isolated also from algae, mollusks, fungi, corals, and bacteria.”

A citation covering this topic was added: [14] Martins et al. 2019. Marine natural flavonoids: chemistry and biological activities. Nat Prod Rev 33, 3260-3272.

Query No. 8: L175 2.2.2. Health Benefits of Common Dietary Flavonoids and Stilbenes via Non-Antioxidant Mechanisms. The referee feels this section should written much compacted, if the conclusion will not be changed!

Response: With all due respect, this subchapter is an essential part of the whole manuscript. What is written in the conclusion is largely based on the data mentioned in this chapter (2.2.2). This subchapter cannot be shortened, since the conclusion would not be backed up by sufficient data. We have thoroughly revisited the subchapter 2.2.2. and came to a conclusion that shortening of this part of manuscript would significantly reduce the quality and relevance of the whole article.

Query No. 9: L232~L234 What is miracule? Because this is scientific review, proncipally the authors depict what is miraculous and negate each ones!

Response: We would like to apologize for the term “miraculous molecules”, it has been deleted, and the sentence now reads as follows: “None of these claims were supported by clinical trials, however.”

Query No. 10: L279~L283 is well done!

Response: Thank you for your comment, we gratefully appreciate it.

Query No. 11: L295~ EGCG should be describes at this position not later!

Response: The term epigallocatechin gallate is firstly used in line 339 and is immediately backed up by abbreviation EGCG, which is then consistently used in the text.

Query No. 12: As for EGCG and catechins, "the only mixture of natural products" which was proved by FDA! The authiors should mention at least, or if the authors know about that, to negate it after depicting.

Shaw T Chen et al

NATURE BIOTECHNOLOGY VOLUME 26 NUMBER 10 OCTOBER 2008

Response: The following sentence was added to the end of this paragraph:“ It is worthy of note that green tea catechins (the so-called sinecatechins) were the first botanical product to be approved as a prescription drug by the FDA”. The above-mentioned reference was added to the text as well as the list of references (No. [64]).

Query No 13: L337 Cameria sinensis should be italic

All plant names are written in roman, theses should be all changes!

L436, L461, L483....

Response: All plant names were in the italic in the original document. We suppose that the names were changed due the process of conversion of the original document to the proof pdf version. All names of botanical species have been revisited and manually converted to italics

Query No. 14: L485 ..iin that the double bond has a Z-configuration, Double bond "does not has a Z-configuration"

Response: With all due respect, this is incorrect. A double bond does have Z- configuration (basic stereochemistry!!!!... in text the prefix was changed to "cis"). 

Query No. 15: It is easy to read, I admit, but L484~L485 is so to apeak wordy!

L506 ....the most common synthetic pathway for

Biosynthetic

Response: This sentence has been divided into two sentences, which read as follows: “As noted above, flavonoids and allied compounds are formed via a combination of cinnamoyl-CoA and three units of malonyl-CoA. This is the most common biosynthetic pathway for the aromatic polyketides.”.

Query No. 16: The refree does not recommend to use "may be" here, for bisnoryangonin derivaties are known to be biosnthesized by Type III

L568 The terminal chain of condensed cinnamoyl-CoA may incorporate a fatty acyl thioester rather

After curcumin topics, L568 is better start by " The terminal chains in plant heptanoids" to explain the topics changed to heptanoids...

Response: The sentence has been changed accordingly: “Some diarylheptanoids are alternatively formed via incorporation of fatty acyl thioester (rather than another cinnamoyl group) to the terminal chain of the cinnamoyl-CoA.” However, there is no change of the topic, heptanoids are already discussed in the paragraph above (curcuminoids).

Query No. 17: L570 Zingiber officinale; italic

Response: This issue is addressed in response of Query No. 13

Query No. 18: The referee considers that shogaol and sanshool are both required to prevent ireus especially after operations on colon cancer. I suppose it is established!

Hydroxy-_ sanshool induces colonic motor activity in rat proximal colon: a possible involvement of KCNK9 Am J Physiol Gastrointest Liver Physiol 308: G579–G590, 2015.

Response: Does the referee really mean hydroxy-α-sanshool? This compound resembles capsaicin (it has fatty acid side chain and incorporated nitrogen in its structure). It is probably formed by the same metabolic pathway as capsaicin (alkaloids), thus it is probably biosynthesized from certain amino acids, and thus has nothing to do with shikimate/aromatic polyketides-derived compounds. We have revisited the publication of Kubota et al, and there is no single mention of shogaol. Due to these reasons, authors apologize to refuse to address this query.

Query No. 19.: The below sentence is inappropriate!

L587 suggests that gingerols and shogaols are not the only active principles of ginger rhizome. Are they also pan-assay interfering compounds?

Response: We agree with the referee, the last sentence of this paragraph has been deleted.

Query No. 20.: L596 italic

Response: This issue is addressed in response of Query No. 13

Query No. 21: L624 believed that isoflavonoids were only found in plants of the Fabaceae family because they were the only plants with the necessary biosynthetic apparatus

What apparatus means?

Response: That means that they were the only plants capable of producing isoflavone synthase (IFS) enzymes, key enzymes in the biosynthesis of isoflavonoids. A parenthesis with the abbreviation IFS was added after this sentence. This abbreviation is used in the sentences above.

Query No. 22: L681~L683 Names should be in italic.

Response: This issue is addressed in response of Query No. 13

L642~L646 Once the authors admitted the role of isoflavonoids in controlling menopausal problems, at last the conclusion sounds it  should not be utilized. The referee feeles that the reader could be confused.    The function of 3"- methylated catechin in hay feber was established. If the authors discuss on curcumin, the micronized formulations curcumins should be handled. because micronization is recently often used, and sometimes it resulted very serious decease by its huge extension of bioavailability of compounds on target. As for esrogenic or non estrogenic activities of isoflavonoids, the referee feels why the authors did not pick up miroesterol in Pueraria mirifica? 

Response:

Regarding isoflavonoids: This is not true. We have not admitted the role of isoflavonoids in controlling menopausal problems, it is clearly stated in the manuscript that: „Isoflavonoids were of interest as agents combating the negative symptoms of menopause such as….”, but: The clinical efficacy of isoflavonoids is still controversial and available data do not support any unequivocal conclusions. Yes, they are available as dietary supplements to treat menopausal symptoms, but that does not necessarily mean, that they are an effective treatment. I think that this is very clear.

Regarding methylated catechin: Following sentences were added to the section dealing with catechins and green tea: “Additionally, methylated catechin (from benifuuki green tea) has been tested as potential anti-allergic agent (e.g. it reduced symptoms of hay fever) [65]. More clinical data are needed to make any reasonable conclusion about this activity..” This is now backed up by reference No. [65]

Regarding curcumin: Following sentences were added to the section dealing with curcumin: “Quite recently, curcumin is also available in the form of micronized powder. This formulation is considered to have increased bioavailability and, therefore, a higher rate of side effects can probably be expected” This is now backed up by reference No. [132].

Regarding miroesterol: Following sentences were added to the section of phytoestrogens: “Another compound that has been largely discussed as a potential phytoestrogen is miroestrol (65, Fig. 15) from Pueraria mirifica (Fabaceae). Miroestrol is available in the form of standardized extracts of P. mirifica and is marketed as a dietary supplement to treat the symptoms of menopause, similarly as other more common isoflavonoids (daidzein, genistein). However, much of miroestrol claims have not been sufficiently proved in clinical trials, and Federal Trade Commission has taken legal action against some marketers of these products.” The structure of was added to the text and is now given under molecule No. 65 in Figure 15

Reviewer 2 Report

Flavonoids and related members of the aromatic polyketide group in human health and disease: Do they really work?”

Authors: Tauchen J, Huml L, Rimpelova S, and Jurasek M

Summary:

Aromatic polyketides, such as flavonoids, have been shown to be promising candidates to treat chronic diseases. This review summarizes the beneficial effects of Aromatic polyketides in clinical trials.

Comments:

A very well researched and written review-article highlighting the great potential and limitations of aromatic polyketides for clinical application.

Minor point:

Line 735-736: The author should be more cautious, when predicting the future potential pharmaceutical applications of natural substances in clinical treatment. Please adjust the sentence.

Page 12, line 459: Please add additional sentence and references: “Additionally, targeting CSCs in CRC enhances CRC chemosensitivity to chemotherapeutic agents such as 5-Fluorouracil, increases apoptosis and supresses migration and metastasis (Buhrmann C et al. 2018 Nutrients; Buhrmann C et al. 2019 Nutrients; Shakibaei M et al. 2014 PLoS One)”. 

Page 15, line 554:  Please add additional reference: Buhrmann  et al. 2014 PLoS One, Toden S et al. 2015 Carcinogenesis.

Author Response

ITEMIZED RESPONSE TO THE REVIEWER’S COMMENTS

Ms. Ref. No.: molecules-884414

Authors: Jan Tauchen, Lukáš Huml, Silvie Rimpelová, Michal Jurášek

Title: Flavonoids and related members of the aromatic polyketide group in human health and disease: Do they really work?

We would like to kindly thank the referee for reviewing our manuscript and suggestions on how to improve it.

A very well researched and written review-article highlighting the great potential and limitations of aromatic polyketides for clinical application.

Minor point:

Query No. 1: Line 735-736: The author should be more cautious, when predicting the future potential pharmaceutical applications of natural substances in clinical treatment. Please adjust the sentence.

Response: We would like to thank you for this suggestion, the last part of the sentence has been removed (i.e. “…and probably will not be pursued as pharmaceutical drugs”).

Query No. 2: Page 12, line 459: Please add additional sentence and references: “Additionally, targeting CSCs in CRC enhances CRC chemosensitivity to chemotherapeutic agents such as 5-Fluorouracil, increases apoptosis and supresses migration and metastasis) (Buhrmann C et al. 2018 Nutrients; Buhrmann C et al. 2019 Nutrients; Shakibaei M et al. 2014 PLoS One)”.

Response: The above-mentioned sentence as well as the references were added to the manuscript.

Query No. 3: Page 15, line 554:  Please add additional reference: Buhrmann  et al. 2014 PLoS One, Toden S et al. 2015 Carcinogenesis.

Response: The above-mention references were added (cited) to the manuscript since we admit its relevance.

Round 2

Reviewer 1 Report

Though Dewick use the word 4-hydroxycinnamate, the referee recommends to use/p/-coumaroyl CoA instead. The reason is the authors does not mention where “4” is at any figure. If the authors suppose the readers are not accustomed to flavonoids who needs so much explains. The 4’ position of flavonoids are the 4 position of hydroxyl group of cinnamate, but the authors did not give numbers on the aromatic ring of from shikimate.

L65~

The referee probably understands the authors intention to write these sentences, however.

L65 There are many “Plant derived aromatic polyketides”, for example famous physiologically active sennoside or hypericin, derived with only acetate units. The authors would like to discharged by using the word “collectively referred”, however. The authors would define as “Aromatic polyketides are a subgroup of compounds derived from shikimate” to deduce flavonoids. But a natural products chemist nowadays knows that flavonoids are one of aromatic polyketides which are formed by various type III PKSs. The referee still wonders why the authors insist to write such a textbook story including some causing misunderstanding on their biosynthesis.

L112 better to write Nringenin-chalcone here instead of chalcone

L113 add Naringenin to chalcone NAringenin chacone is better probably.

Because there is no explanation how isoliquiritigenin forms, yet there is in the figure #1 structure. But the authors do not use the structure anywhere. The best way is cut the structure from the figure. Probably the reason why the authors added the liquiritigenin there is that it is the precursor of many isoflavonoids. But the authors did not mention of loss of 5^th position hydroxyl group. The major flavonoids come from naringenin-chalcone. If the authors think that the reduction or dehydration occurs after chalcone formation, it is simply wrong. The reductase works with polyketide forming with chalcone synthase concomitantly but without associated style, which was proved by JP Noel usingsemipermeable membrane more than 10 years ago. Once chalcone forms or the ketone groups aromatized, then it is very hard to be reduced. The aromatization should be considered completely different from enolaization. The authors at least should mention/p/‐coumarylcyclohexantrione, which is a proposed CHR substrate by JPNoel.

L113 Naringenine-Chalcones can then undergo enolization or

L114_dehydration/enolization_and Michael-type nucleophilic attack of a phenol OH-group onto the α,β-

L115 unsaturated ketone (chalcone isomerase), forming a general 15-carbon flavonoid structure consisting

L116 of two phenyl rings (A and B) and a six-membered heterocyclic ring (C), although 116 not all flavonoids

L117 possess the heterocyclic ring) [6–8].

  * L417 Just comment, the authors may know, Rohitukine (30, Fig. 7) is
    probably aromatic polyketides, yet the belows say that endphytoc!
  * Phytomedicine: international journal of phytotherapy and
    phytopharmacology 21(4) DOI:10.1016/j.phymed.2013.09.019
  * L485 p-coumaroyle CoA could be better here because most flavonoids
    use coumaroyl-CoA but limited ones utilize cinnamoyl-CoA, for
    example/Scutellaria/
  * 485 As noted above, flavonoids and allied compounds are formed via a
    combination of cinnamoyl-
  * 486 CoA and three units of malonyl-CoA.

L528 and L551 cinnamoyl- should be to coumaroyl- because DCS or gingerol synthase would not accept cinnamoyl CoA.

We can use the word cinnamate as a broader mean of biosynthetic precursor , but cinnamate should be oxidized before turning to CoA derivative.

In ginger,

    Transcriptome Analysis Provides Insights into Gingerol Biosynthesis
    in Ginger(/Zingiber officinale///)

<https://acsess.onlinelibrary.wiley.com/doi/10.3835/plantgenome2018.06.0034>

*https://doi.org/10.3835/plantgenome2018.06.0034*

605 believed that isoflavonoids were only found in plants of the Fabaceae familybecause they were the

606 only plants with the necessary biosynthetic apparatus (i.e. IFS enzymes)

Author Response

Response to the Reviewer 1 (round 2) on the manuscript 884414 (Molecules)

Title:

Flavonoids and related members of the aromatic polyketide group in human health and disease: Do they really work

Authors:

Jan Tauchen, Lukáš Huml, Silvie Rimpelová, Michal Jurášek

Dear Sir/Madam,

We would like to thank the reviewer for thorough reading and reviewing of our manuscript and especially for her/his remarks that helped us to improve our manuscript. We have taken the reviewer’s advice and comments into account carefully point-by-point and made the following changes and corrections in the manuscript.

Reviewer 1:

Comment 1:

Though Dewick use the word 4-hydroxycinnamate, the referee recommends to use/p/-coumaroyl CoA instead. The reason is the authors does not mention where “4” is at any figure. If the authors suppose the readers are not accustomed to flavonoids who needs so much explains. The 4’ position of flavonoids are the 4 position of hydroxyl group of cinnamate, but the authors did not give numbers on the aromatic ring of from shikimate?

Changes, corrections and explanations in the revised manuscript regarding Comment 1:

Thank you very much for this comment. The term 4-hydroxycinnamoyl-CoA was changed to p-coumaroyl-CoA in all parts of the text where this name occurred (see other comments).

Comment 2:

The referee probably understands the authors intention to write these sentences, however.

L65 There are many “Plant derived aromatic polyketides”, for example famous physiologically active sennoside or hypericin, derived with only acetate units. The authors would like to discharged by using the word “collectively referred”, however. The authors would define as “Aromatic polyketides are a subgroup of compounds derived from shikimate” to deduce flavonoids. But a natural products chemist nowadays knows that flavonoids are one of aromatic polyketides which are formed by various type III PKSs. The referee still wonders why the authors insist to write such a textbook story including some causing misunderstanding on their biosynthesis.

Changes, corrections and explanations in the revised manuscript regarding Comment 2:

We would like to thank the reviewer for this suggestion. The following section enlightening the problematics of flavonoid nomenclature and what makes them different from other polyketides with aromatic functional groups, such as sennosides and hypericin, has been added to the text (L67-72):

“There are other plant-derived polyketides with aromatic functional group (such as emodin, sennosides, and hypericin). However, in the course of biosynthesis these structures utilize acetate starter units (e.g. acetyl-CoA, propionyl-CoA) instead of that derived from shikimate metabolism (e.g. p-coumaroyl-CoA) and are usually products of type II polyketide synthase (PKS) enzymes (not type III PKS; see further sections for more details).”

Comment 3:

L112 better to write Nringenin-chalcone here instead of chalcone

Changes, corrections and explanations in the revised manuscript regarding Comment 3:

We would like to thank the reviewer for this suggestion, the term chalcone was adjusted to naringenin chalcone.

Comment 4:

L113 add Naringenin to chalcone NAringenin chacone is better probably.

Changes, corrections and explanations in the revised manuscript regarding Comment 4:

We would like to thank the reviewer for this suggestion, the term chalcones was adjusted to naringenin chalcone.

Comment 5:

Because there is no explanation how isoliquiritigenin forms, yet there is in the figure #1 structure. But the authors do not use the structure anywhere. The best way is cut the structure from the figure. Probably the reason why the authors added the liquiritigenin there is that it is the precursor of many isoflavonoids. But the authors did not mention of loss of 5^th position hydroxyl group. The major flavonoids come from naringenin-chalcone. If the authors think that the reduction or dehydration occurs after chalcone formation, it is simply wrong. The reductase works with polyketide forming with chalcone synthase concomitantly but without associated style, which was proved by JP Noel usingsemipermeable membrane more than 10 years ago. Once chalcone forms or the ketone groups aromatized, then it is very hard to be reduced. The aromatization should be considered completely different from enolaization. The authors at least should mention/p/‐coumarylcyclohexantrione, which is a proposed CHR substrate by JPNoel.

Changes, corrections and explanations in the revised manuscript regarding Comment 5:

We would like to thank the reviewer for this comment. Whole section 2. Flavonoids and stilbenes have been changed according to these comments. Because of these changes, a structure of liquiritigenin is still present in the manuscript.

Comment 6:

L113 Naringenine-Chalcones can then undergo enolization or.

Changes, corrections and explanations in the revised manuscript regarding Comment 6:

This comment was reflected already in comment 4.

Comment 7:

L114_dehydration/enolization_and Michael-type nucleophilic attack of a phenol OH-group onto the α,β-.

L115 unsaturated ketone (chalcone isomerase), forming a general 15-carbon flavonoid structure consisting.

L116 of two phenyl rings (A and B) and a six-membered heterocyclic ring (C), although 116 not all flavonoids.

L117 possess the heterocyclic ring) [6–8].

Changes, corrections and explanations in the revised manuscript regarding Comment 7:

The sentence was changed accordingly: “Naringenin-chalcone can then undergo enolization and Michael-type nucleophilic attack of OH-group onto the α,β-unsaturated ketone (through chalcone isomerase), forming a general 15-carbon flavonoid structure consisting of two phenyl rings (A and B) and a six-membered heterocyclic ring (C)”

Comment 8:

  * L417 Just comment, the authors may know, Rohitukine (30, Fig. 7) is

    probably aromatic polyketides, yet the belows say that endphytoc!

* Phytomedicine: international journal of phytotherapy and

    phytopharmacology 21(4) DOI:10.1016/j.phymed.2013.09.019

Changes, corrections and explanations in the revised manuscript regarding Comment 8:

We appreciate this comment, many thanks. Following sentences have been added to the text: “It was initially thought that rohitukine is plant-derived. However, it has now been shown that rohitukine is probably not biosynthesized by the plant itself, but is the product of an endophytic fungi.”

Comment 9:

    * L485 p-coumaroyle CoA could be better here because most flavonoids

    use coumaroyl-CoA but limited ones utilize cinnamoyl-CoA, for

    example/Scutellaria/

  * 485 As noted above, flavonoids and allied compounds are formed via a

    combination of cinnamoyl-

  * 486 CoA and three units of malonyl-CoA.

Changes, corrections and explanations in the revised manuscript regarding Comment 9:

The term cinnamoyl-CoA was corrected for p-coumaroyl-CoA

Comment 10:

  L528 and L551 cinnamoyl- should be to coumaroyl- because DCS or gingerol synthase would not accept cinnamoyl CoA.

We can use the word cinnamate as a broader mean of biosynthetic precursor , but cinnamate should be oxidized before turning to CoA derivative.

In ginger,

    Transcriptome Analysis Provides Insights into Gingerol Biosynthesis

    in Ginger(/Zingiber officinale///)

<https://acsess.onlinelibrary.wiley.com/doi/10.3835/plantgenome2018.06.0034>

*https://doi.org/10.3835/plantgenome2018.06.0034*

  605 believed that isoflavonoids were only found in plants of the Fabaceae familybecause they were the

606 only plants with the necessary biosynthetic apparatus (i.e. IFS enzymes).

Changes, corrections and explanations in the revised manuscript regarding Comment 10:

Cinnamoyl was changed to coumaroyl in both lines as suggested. The sentence commenting the restriction of isoflavonoids to Fabaceae family has been changed: “Initially it was believed that only the leguminous plants do possess the necessary biosynthetic apparatus (i.e. IFS enzymes) for biosynthesis of isoflavonoids, and thus that these compounds are entirely limited to members of Fabaceae family.”